# Persistence of plant-mediated microbial soil legacy effects in soil and inside roots

S. Emilia Hannula [1,5 ✉], Robin Heinen [1,2,5], Martine Huberty[1,3], Katja Steinauer[1,4], Jonathan R. De Long[1], Renske Jongen[1] & T. Martijn Bezemer [1,3]

Plant-soil feedbacks are shaped by microbial legacies that plants leave in the soil. We tested the persistence of these legacies after subsequent colonization by the same or other plant species using 6 typical grassland plant species. Soil fungal legacies were detectable for months, but the current plant effect on fungi amplified in time. By contrast, in bacterial communities, legacies faded away rapidly and bacteria communities were influenced strongly by the current plant. However, both fungal and bacterial legacies were conserved inside the roots of the current plant species and their composition significantly correlated with plant growth. Hence, microbial soil legacies present at the time of plant establishment play a vital role in shaping plant growth even when these legacies have faded away in the soil due the growth of the current plant species. We conclude that soil microbiome legacies are reversible and versatile, but that they can create plant-soil feedbacks via altering the endophytic community acquired during early ontogeny.

[1] Netherlands Institute of Ecology, Department of Terrestrial Ecology, Droevendaalsesteeg 10, 6708 PB Wageningen, The Netherlands. [2] Technische Universität München, Wissenschaftszentrum Weihenstephan für Ernährung, Landnutzung und Umwelt, Lehrstuhl für Terrestrische Ökologie, Hans-Carl-von-Carlowitz-Platz 2, 85354 Freising, Germany. [3] Institute of Biology Leiden, Plant Ecology and Phytochemistry, Sylviusweg 72, 2333 BE Leiden, The Netherlands. [4] Present address: Institute of Plant Sciences, University of Bern, Altenbergrain 21, 3013 Bern, Switzerland. [5] These authors contributed equally: S. Emilia Hannula, Robin Heinen. ✉email: e.hannula@nioo.knaw.nl

Soil microbes are widely acknowledged to be major drivers of plant growth and plant community assembly[1]. Plants affect soil microbes via the quantity and quality of rhizodeposits[2,3], and litter[4]. These plant-mediated changes in the soil microbiome can influence the growth of other plant species that grow later in the same soil (plant-soil feedbacks[5–7]). Plants can negatively influence succeeding plant species through accumulation of pathogenic microbes in the soil[7–10], or positively through the build-up of beneficial or mutualistic microbes[10–12]. These microbiome-mediated plant-soil feedbacks may be general among functional groups of plants (e.g., grasses and forbs) as these groups markedly differ in their effects on - and sensitivity to - soils. While microbial soil legacies can have major impacts on plant growth[2], we are still far from understanding and predicting these legacy effects. Specifically, we do not know how persistent soil legacies are (i.e., how long they last after the removal of the plant), and if these microbial legacy effects vary between different plant species that both shape and respond to soil legacies[13,14].

While a 'current plant' grows in soil conditioned by a 'previous plant', it will respond to biotic and abiotic soil conditions, but simultaneously change the microbial legacy in the soil. How these temporal changes contribute to the overall outcome of plant-soil feedbacks is not well understood. The specific influence of the previous plant on the soil community is a widely held assumption behind plant-soil feedback theories and experiments, whereas the effect of the newly created soil legacies influenced by the current plant and the combination of the two types of legacies, is often overlooked and lacks rigorous empirical testing. The sensitivity of a plant to the soil microbial community may vary depending on the age of the plant[15–17] and generally, seedlings are considered to be more sensitive to for example pathogen effects than adult plants[18]. Besides having an inherited seed microbiome from its parental plants[19], freshly germinated seedlings can experience only the soil legacy of the previous plant, whereas older plants will experience soils that bear a legacy of previous plants, but which may have also been modified by themselves. Interestingly, a recent study proposed that the soil microbial community present at the plant germination stage may be a stronger determinant of plant growth than the soil microbial communities that are present at later ontogenetic plant stages[20]. Moreover, studies suggest that seedlings are more susceptible to endophytes (i.e. microbes living inside plant roots) colonizing the roots than adult plants[21], which may be due to the low levels of chemical defenses in younger plants[19] or their greater need for symbiotic partners to survive.

Endophytic microbes living inside the roots are in closer contact with the plant than the microbes in the soil[22]. Multicellular fungi may simultaneously grow hyphae in rhizosphere soil and in the endosphere, whereas unicellular bacteria cannot, which may lead to differences between the two microbial kingdoms[23,24]. Endophytes can be beneficial for plant growth through their effects on plant nutrient status, through the protection they provide against pathogens and pests, and via increasing stress tolerance and modulation of plant development[23–27]. Plants inherit endophytic microbes through transfer of microbes from parental plants in seeds[19] but also select their own endophytic microbes from the pool available in the soil[24,27] and as such, the community structure of endophytes within a plant species is known to differ between soils with a different history[28,29]. The plant-mediated legacy of a soil may thus affect the endophytes the plant acquires which, in turn, can affect plant growth and performance[30]. Yet, it is largely unclear how the identity of the previous plant and plant traits affect the composition of endophytes across plant species. As many endophytes are acquired at early growth stages and often remain in the plant throughout its growth, this suggests that exposure to soil legacies of a previous plant early in life can have long-lasting effects on plant growth, even when these legacies are no longer detectable in the soil surrounding the plant root.

To examine the persistence of plant-specific soil microbial legacies during the next generation of plant growth, we set up a long-term mesocosm common garden experiment with six plant species, belonging to grasses and forbs, that are commonly found in former agricultural grasslands, all known to form a symbiosis with arbuscular mycorrhizal fungi (AMF) and of which plant-soil feedbacks have been well-studied over the past two decades[31,32]. As grass species all belong to the same family, but forbs do not, we selected forb species from one family as well to make our design phylogenetically balanced. The three selected grasses were *Alopecurus pratensis*, *Festuca ovina*, and *Holcus lanatus* (all Poaceae) and the three selected forbs were *Hypochaeris radicata*, *Jacobaea vulgaris*, and *Taraxacum officinale* (all Asteraceae). We first created six distinct soil microbial legacies by growing the plants as monocultures in 200-L soil mesocosms for 12 months[33]. We then divided each mesocosm into six physically separated sections (by placing soil in buckets), in which we planted all the six responding plant species (see Fig. 1 for set-up). We monitored the soil microbiome in each section by non-destructive repeated sampling for five months and examined changes in the microbiome (bacteria and fungi) caused by the previous and the current plant over time. After 5 months of plant growth, we destructively harvested the plants to examine their responses to the soil legacies and analyzed the root endophytic microbiome[22,29].

Plants create directional changes in soil microbiomes that differ between plant species and their functional group[33–35]. Previous work has shown that bacterial soil legacies have a faster turnover time than fungal legacies[33,36] due to differences in turnover rates and traits related to microbial growth strategies[37]. Following these general assumptions, we tested the following hypotheses:

(i) The soil microbial legacy of the previous plant will diminish over time, while the effects of the current plant on the soil microbial community will concomitantly increase with time (Fig. 2). Furthermore, we expect that the legacy effects will be detected for longer time periods in the fungal than in the bacterial community.

(ii) As the endophytes are acquired by plants in early growth stages, the legacy effect of the previous plant, even though it may not be detected anymore in the soil, will still be visible in the endophytic root microbiomes of the current plant.

(iii) Endophytic microbes will have stronger relationships with plant growth than rhizosphere microbes.

(iv) Conspecific feedbacks are due to an accumulation of specific communities, microbial groups, functional guilds (such as mutualists or pathogens), or microbial species in the endosphere and rhizosphere

To test these hypotheses, we analyzed both bacterial and fungal communities in the soils in the beginning of the experiment and at three time points during plant growth and inside the roots at final harvest, and relate the community composition of microbes in soils conditioned by different previous plant species and families to the plant biomass of the current plant at the final harvest.

We show that the legacy of the previous plant stays lingering in the soil fungal community while in soil bacterial community the memory fades away quickly. Yet, both legacies are stored inside the plant roots and affect the growth of the following plant.

## Results
**Directional changes in soil microbiomes.** First, we investigated the direction of temporal changes in microbiomes due to current and previous plant species and plant families (for expectations see

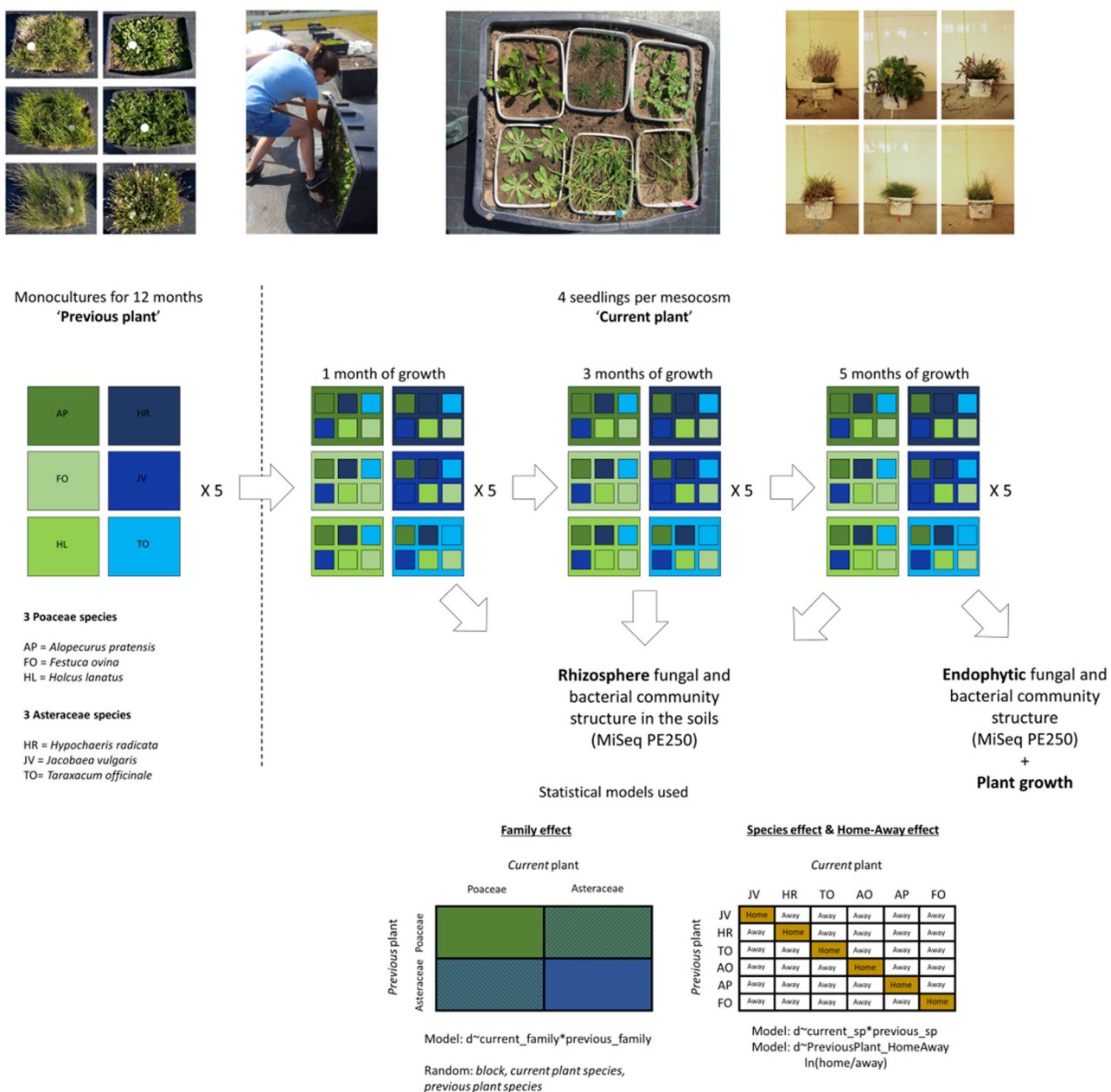

**Fig. 1 Set-up of the experiment.** In short, monocultures of six plant species were maintained for 12 months in 30 (6 species × 5 replicates) mesososms. Then, the top soil was divided into six smaller containers and placed back into the original containers (in mid-May) See Supplementary Movie 1 for how this was done. The existing plants were removed and after three weeks (in June) four seedlings of one of the same six plant species were planted in each of the six smaller containers reciprocally. This equaled to (30 mesocosms × 6 plants) 180 smaller mesocosms. Soil from each of these smaller mesocosms were sampled one month (July), three months (September) and five months (November) after planting the plants. Soil was sampled so that four soil cores were taken per mesocosm, one sample next to each plant and combined into one composite sample per small mesocosm. After five months, the containers were destructively harvested, plant biomass was measured from dried and washed root and shoot material, and endophytic microbial communities were surveyed from sterilized root samples. Statistical models used for the evaluation of current and previous plant family and the effect of growing in own soil (home-away) effects. Same models were used for both plant and microbial data. See text for details.

Fig. 2). After one year of plant growth, each plant species had created its unique microbiome (PerMANOVA $R^2 = 0.35$ for fungi and $R^2 = 0.24$ for bacteria, for both $p < 0.001$; Fig. 3A, C) and the soil microbiomes also differed significantly between the two plant families (PerMANOVA $R^2 = 0.13$ for fungi and $R^2 = 0.07$ for bacteria, for both $p < 0.001$; Figs. 3B, D and Supplementary Figs. 1 and 2). The effects of the previous and current plant family on bacterial and fungal communities in the soil changed over time and the pattern partly followed hypothesis (i) that the effect of the current plant species would in time outweigh the effect of the

previous plant species. Specifically, 1 month after planting the current species, for both the fungi and bacteria, a larger part of the variation in the soil microbial community structure was explained by the identity of the previous plant than by that of the current plant (for bacteria: PerMANOVA previous $R^2 = 0.09$ and current $R^2 = 0.05$, for both $p < 0.001$ and for fungi PerMANOVA previous $R^2 = 0.11$, $p < 0.001$ and current $R^2 = 0.03$, $p = 0.11$; Fig. 3A, B and Supplementary Figs. 1–3). For soil bacteria, the effects of the previous plant diminished over time, while the effects of the current plant increased (Fig. 3C). For soil fungi,

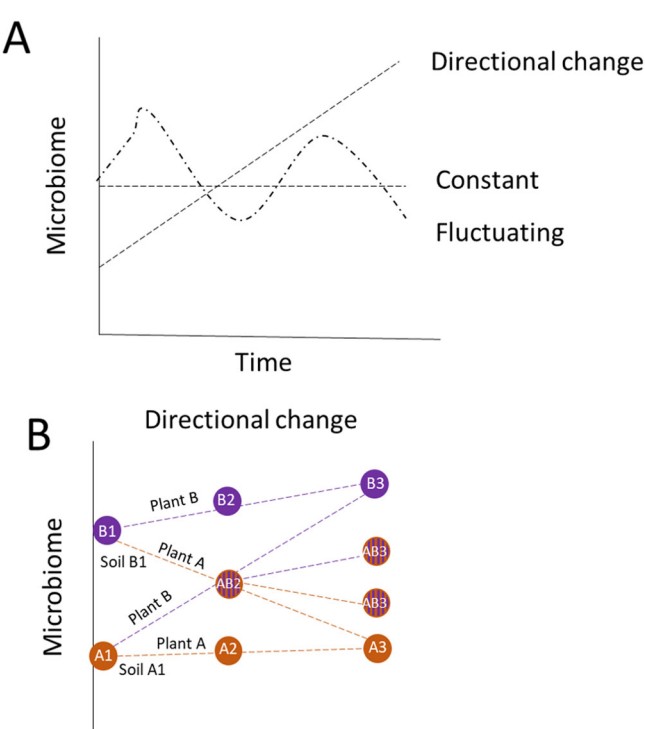

**Fig. 2 Theoretical change in microbiomes over time.** Theoretical framework of the effects of conditioning (previous) and responding (current) plants on soil microbial community composition. **A** we expect that the microbiome will change in time under a new plant community in either a directional, constant, or fluctuating way. **B** Furthermore, if this change is directional, we expect that the change will lead to convergence in microbiomes and that over time the microbiome will develop into a species-specific microbiome type but that this will depend on the initial microbiome composition and hence on the legacy of the plant that grew in the soil before.

however, the amount of variation in community composition that was explained by the legacy of the previous plant species was highest 3 months after planting the new species (PerMANOVA $R^2 = 0.13$, $p < 0.001$), and 5 months after planting, this effect was still present and larger than the amount of variation explained by the current plant species (PerMANOVA $R^2 = 0.11$, $p < 0.001$; Fig. 3A).

Interestingly, especially for fungi, the family the previous and current plant belonged to had a large effect on community assembly. Poaceae and Asteraceae left distinct fungal legacies and also responded differently to family legacies (Fig. 3B and Supplementary Fig. 3). One month after planting the current plants, most of the variation in fungal communities in the soil was explained by whether the plant was growing in soil with a legacy of a Poaceae or of an Asteraceae (PerMANOVA $R^2 = 0.08$, $p < 0.001$; ANOVA on NMDS1: $F_{1,34} = 41.75$, $p < 0.001$, Fig. 3B and Supplementary Figs. 1 and 3). Three months after planting the current plants, we detected a significant effect of the current plant family (PerMANOVA $R^2 = 0.05$; ANOVA on NMDS1: $F_{1,34} = 97.60$, for both $p < 0.001$), but also of the family the previous plant belonged to (PerMANOVA $R^2 = 0.08$; ANOVA on NMDS1: $F_{1,34} = 18.71$, for both $p < 0.001$). Each current/previous family combination created a unique mycobiome type (ANOVA on NMDS1: $F_{3,32} = 38.84$, $p < 0.001$; Fig. 3B) so that soils with Asteraceae currently growing in them started to become more similar to soils with a legacy of Asteraceae, while soils with Poaceae currently growing in them moved towards soils with a

Poaceae legacy. After 5 months of growth, we could distinguish three (out of four possible combinations of previous and current plant families) distinct mycobiomes. Soils with an Asteraceae legacy in which currently a Poaceae was grown and soils with a Poaceae legacy in which currently an Asteraceae was grown had changed to such an extent that they became very similar to each other (ANOVA on NMDS1: $F_{3,32} = 26.42$, $p < 0.001$; Fig. 3B).

For bacteria, the soil microbiomes of the two families differed less, and for the first two time points, no effect of previous or current family was detected (Fig. 3D). However, 5 months after planting the current plants, bacterial communities differed between soils in which currently a Poaceae or an Asteraceae was grown, but only in the soils with a legacy of Poaceae (LME on NMDS1; identity of current and precious plant as random factors: $F_{3,32} = 26.42$, $p < 0.001$; Fig. 3D). Importantly, the time point of sampling affected the bacterial community composition much more strongly (PerMANOVA $R^2 = 0.39$, $p < 0.001$; Supplementary Fig. 2) than the fungal community composition (PerMANOVA $R^2 = 0.09$, $p < 0.001$; Supplementary Fig. 1) indicating that the bacterial community is less temporally stable than the fungal community ('fluctuating' in Fig. 2A).

**Effect of soil and its microbiome on plant growth**. We destructively harvested the current plants 5 months after planting them, to investigate if plant growth was affected by previous plant-mediated soil legacies. Aboveground biomass of all plant species was affected by the soil legacies caused by the previous plant (Fig. 4A, B). For Poaceae, aboveground biomass in soils with a legacy of any Poaceae species was lower than in soils with an Asteraceae legacy (ANOVA $F_{1,24} = 19.52$, $p < 0.001$; Fig. 4A), and for the three Asteraceae species, a negative effect of growing in their "own" species-specific soil was detected (Fig. 4A). Across all plant species, aboveground biomass at harvest was related to the soil fungal community structure at all the time points measured (ENVFIT: $R^2 = 0.04–0.06$, $p < 0.01$; Fig. 5A), while no relationship was observed between the bacterial community structure in the soil at any measuring time point and aboveground plant biomass at harvest (Fig. 5B). Correlations between soil fungal community structures and plant shoot biomass at harvest were strongest after 5 months of plant growth (ENVFIT: $R^2 = 0.06$, $p < 0.001$; Fig. 5C) which was related also to the first axis of a multivariate NMDS ordination for the fungal community (Pearson: $R^2 = 0.07$, $p < 0.001$). When the effects of the soil fungal community structure after 5 months on plant biomass at harvest were evaluated for each test plant species separately, we detected that the aboveground biomass of all Poaceae species was explained by soil fungal community composition, but for the Asteraceae, only the shoot biomass of *Taraxacum officinale* (TO) was weakly related to soil fungal community composition (Fig. 5D).

**Legacy effects on endophytic microbes**. To evaluate hypothesis (ii) that the previous plant influences the endophytic root microbiome of the current plant, we examined the effect of the previous and current plant species on the fungal and bacterial communities inside plant roots. The two families to which the previous plant belonged significantly differed in how they influenced endophytic fungi (Fig. 6A) and endophytic bacteria (Fig. 6B) of the current plant. Furthermore, there was an interaction between the family of the previous plant and the family of the current plant on endophytic community structures (Fig. 6A, B). For both endophytic bacteria and fungi, the identity of the current plant explained most of the variation in community structure (PerMANOVA $R^2 = 0.48$ for bacteria and $R^2 = 0.26$ for fungi, for both $p < 0.001$; Fig. 6C). However, the legacy of the previous plant also significantly affected the composition of the bacterial and fungal root endophytic communities

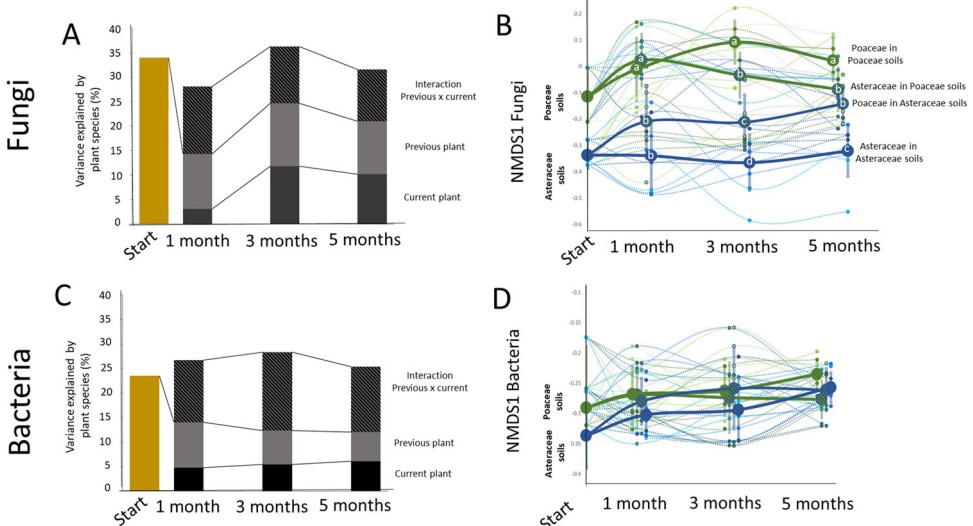

**Fig. 3 Effects of current and previous plants and their family on soil microbiomes.** Variance in fungal (**A**) and bacterial (**C**) community structure in soils explained by current and previous plant species and their interaction in the beginning (explained by the then current but later referred as previous plant species) of the experiment, and after 1, 3, and 5 months. Changes in community structure of fungi (**B**; depicted as NMDS1, for full data see supplementary figure 1) and of bacteria (**D**; depicted as NMDS1, for full data see supplementary fig. 2) in time and per plant species (thin lines, $n = 5$) and averaged per plant functional group (thick lines, $n = 45$) are presented. The colors refer to plant species, green colors mark Poaceae and blue colors Asteraceae. The circles note the average value for NMDS1 for each combination of previous and current plant families ($n = 30$) and error bars of points on the thick line depict the standard error between plant species from same families. The letters in the circles note statistical significance of the Tukey post doc test between combinations of families of current and previous plant estimated with lme model where the identity of current and precious plant was used as random factors (Fig. 1). The 2D stress values of the across time points ordinations were 0.19 for fungi and 0.21 for bacteria.

(PerMANOVA $R^2 R^2 = 0.06$ for bacteria and $R^2 = 0.09$ for fungi, for both $p < 0.001$; Fig. 6C), and there was a significant interaction between the legacy of the previous plant and the identity of the current plant ($R^2 = 0.09$ and $R^2 = 0.15$, for bacteria and fungi, for both $p < 0.005$; Fig. 6C). Although significant, the effect sizes of plant family on soil and endophytic microbiomes after 5 months of growth were rather small; the effect size of the previous plant family was larger in the roots than in the soil for fungi ($R^2 = 0.09$ in the roots and $R^2 = 0.06$ in the soil), and for bacteria ($R^2 = 0.07$ in the roots and $R^2 = 0.02$ in the soil). When root microbiome was compared with soil microbiome we detected that fungal community in roots resembled most the fungal community in soils sampled at the same time as roots after 5 months of plant growth ($R^2 = 0.19$, $p < 0.001$) while for bacteria the resemblance was weaker and the strongest correlation was detected after 3 months of plant growth which is 2 months before sampling the endophytes ($R^2 = 0.03$, $p < 0.05$; Supplementary Fig. 4). Different endophytic fungal and bacterial groups were selected for by current plant species that were growing in soil with a legacy of previous plant species (Fig. S5) Bacterial phyla like *Actinobacteria*, *Patescibacteria*, and *Fibrobacteres* were strongly affected by current plant species (LME: $F$ values > 20, $p < 0.001$ after FDR correction), while especially fungal classes such as Magnaporthales and Sebacinales and also bacterial phyla such as *Acidobacteria* and *Nitrospirae* were affected interactively by both the previous and the current plant (LME: $F$ values for previous plant >5, $p < 0.001$ after FDR correction; Supplementary Fig. 5).

We subsequently related the endophytic community composition to the plant growth parameters. Root biomass was significantly related to bacterial community composition inside the roots (ENVFIT: $R^2 = 0.05$, $p < 0.05$; Fig. 6E). Specifically, the relative abundance of *Actinobacteria* and *Patescibacteria* inside the roots correlated with greater root biomass (Fig. 6F and Supplementary Fig. 6). For all but one of the current plant species, there was a significant relationship between endophytic bacterial community and

root biomass, but the magnitude of the effect (measured as $R^2$) varied among species (Supplementary Fig. 7). For fungi, the relative abundance of arbuscular mycorrhizal fungi inside the roots was associated with root biomass across plant species and this relationship was negative (Pearson: $R^2 = 0.06$, $p < 0.01$; Fig. 6D and Supplementary Fig. 6).

**Microbiome effects on plant growth in own soils**. Lastly, we tested hypothesis (iv) on the role of microbiome, microbial groups, and individual microbes in regulation of conspecific feedback, i.e., plants growing on soil legacies created by their own species. On average, all plant species used in this experiment performed worse in soils with a legacy of the same plant species than in soils from other plant species (Figs. 4B and 7A). For three plant species (*Holcus lanatus*, *Jacobaea vulgaris*, and *Taraxacum officinale*) fungal community was significantly different when plants were grown in the soils where the same plant species had grown earlier than in soils from other plant species 1 month after planting (PERMANOVA: $R^2 > 0.05$, $p < 0.01$; Fig. 7B and Supplementary Fig. 8). For *Holcus lanatus*, this effect was related to the larger relative abundance of fungal plant pathogens when grown in its own soil but only at the first time point ($F = 8.63$, $p = 0.006$: Supplementary Fig. 10), while for *Jacobaea vulgaris* we detected a reduction in the relative abundance of AMF when it was grown in its own soil compared to other soils at the first time point ($F = 6.65$, $p = 0.016$: Supplementary Fig. 10). For the other species (*Alopecurus pratensis*, *Festuca ovina*, and *Hypochaeris radicata*), a significant effect on fungal communities of growing in their own soils was observed in the soil samples collected at 5 months (Fig. 7B and Supplementary Fig. 8). Soil samples of *J. vulgaris*, *F. ovina*, and *H. radicata*, had a significantly different bacterial composition when these species were grown in their own soil after three and 5 months of growth (Fig. 7B and Supplementary Fig. 9). *Alopecurus pratense*, *F. ovina*, and *J. vulgaris*, had a significantly different endophytic fungal community structure when

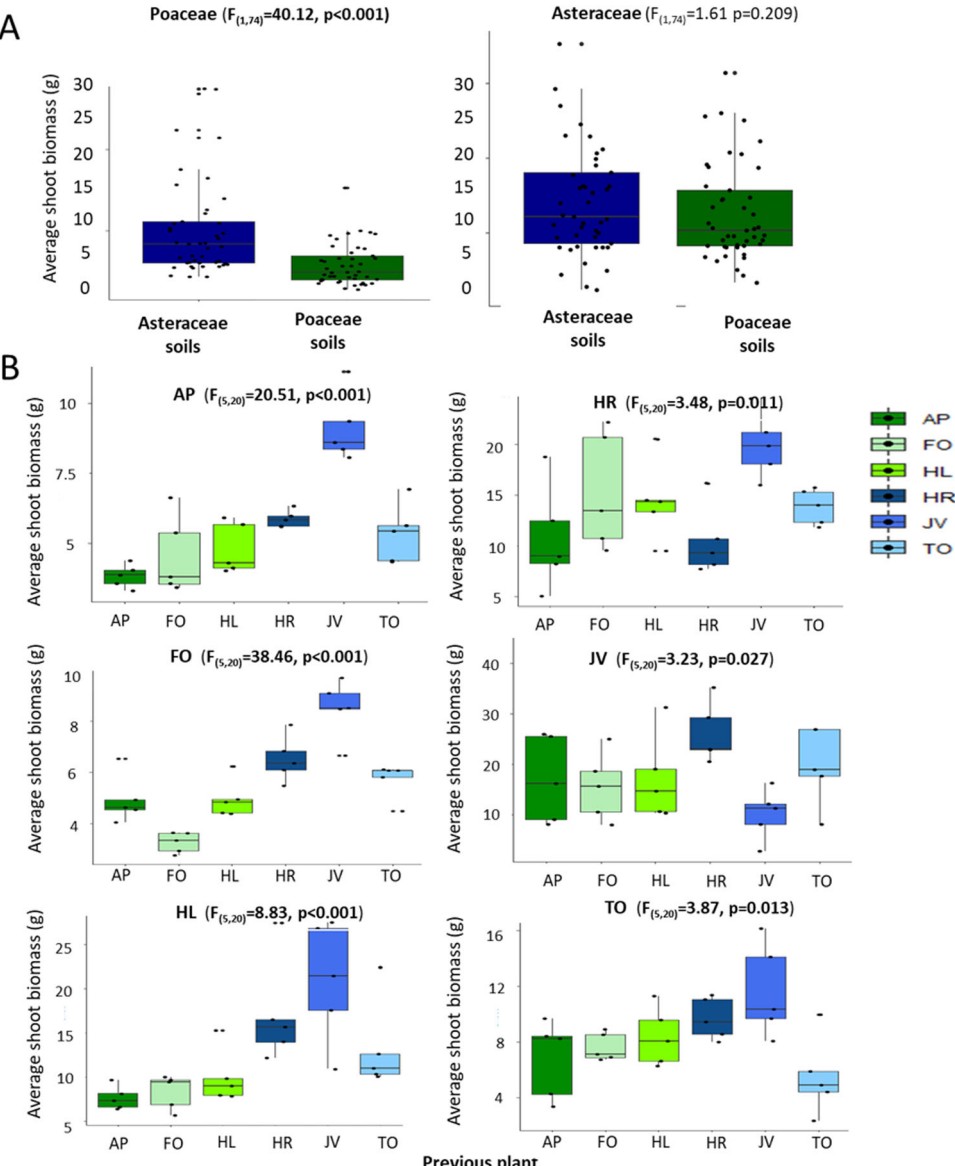

**Fig. 4 Plant growth across soils.** Plant growth in the feedback phase affected by the previous plant and explained by soil microbial communities. Average shoot biomass (dw) of the responding Poaceae and Asteraceae grown in Poaceae and Asteraceae soils (**A**) and of of individual plants grown in all the possible soils (**B**). The Tukey box-and-whisker-plots depict median shoot biomass of plants and individual points show the variation between replicate plants ($n = 45$ for **A** and $n = 5$ for **B**). Poaceae are depicted with green colors and Asteraceae with blue. The abbreviations for plant species are *AP Alopecurus pratensis*, FO *Festuca ovina*, HL *Holcus lanatus*, HR *Hypochaeris radicata*, JV *Jacobaea vulgaris*, TO *Taraxacum officinale*. Statistical values are shown above each graph for the full model (lme model shown in Fig. 1).

grown in their own soil, while *A. pratense*, *F. ovina*, and *H. radicata*, had a significantly different endophytic bacterial composition when grown in their own soil (Fig. 7B, 'roots').

Due to the importance of endophytes for plant performance we further investigated for the endophytic compartment, which microbial groups changed when the current plant was grown in conspecific soil, or in soils from a plant from the same family. We detected that the relative abundance of potential fungal plant pathogens in the roots of all Poaceae plants increased when grown in their respective own soil (log-transformed relative abundance LME: $F = 13.84$, $p < 0.001$; Fig. 7C and Supplementary Fig. 10). Asteraceae growing in Poaceae soils exhibited a higher relative abundance of plant pathogens and this was independent from whether the plants were grown in their own soil or in another soil (log-transformed relative abundance LME: $F = 21.41$,

$p < 0.001$, Supplementary Fig. 10). For thePoaceae species, different plant pathogens were enriched in roots when grown in their own soil (Supplementary Fig. 11) compared to growing in soils with a legacy of other Poaceae species. The main pathogens that were enriched when grown in own soil compared to other soils were for *A. pratensis*, Magnaporthiopsis species, for *F. ovina, Alternaria* sp. and *Slopeiomyces cylindrosporus* and for *H. lanatus*, Magnaporthiopsis species and *Neoerysiphe nevoi*. We did not detect strong positive selection in roots of plants grown in their own soils for AMF and only some saprotrophic taxa such as Orbiliaceae, *Ceratobasidium* sp. and *Marasmius* sp. were affected by plants growing in their own soils (Supplementary Fig. 11). Furthermore, two fungal orders, Agaricales (LME: $F = 5.50$, $p = 0.020$) and Magnaporthales (LME: $F = 9.13$, $p = 0.003$), were enriched in roots of Poaceae species that grew in their own soil,

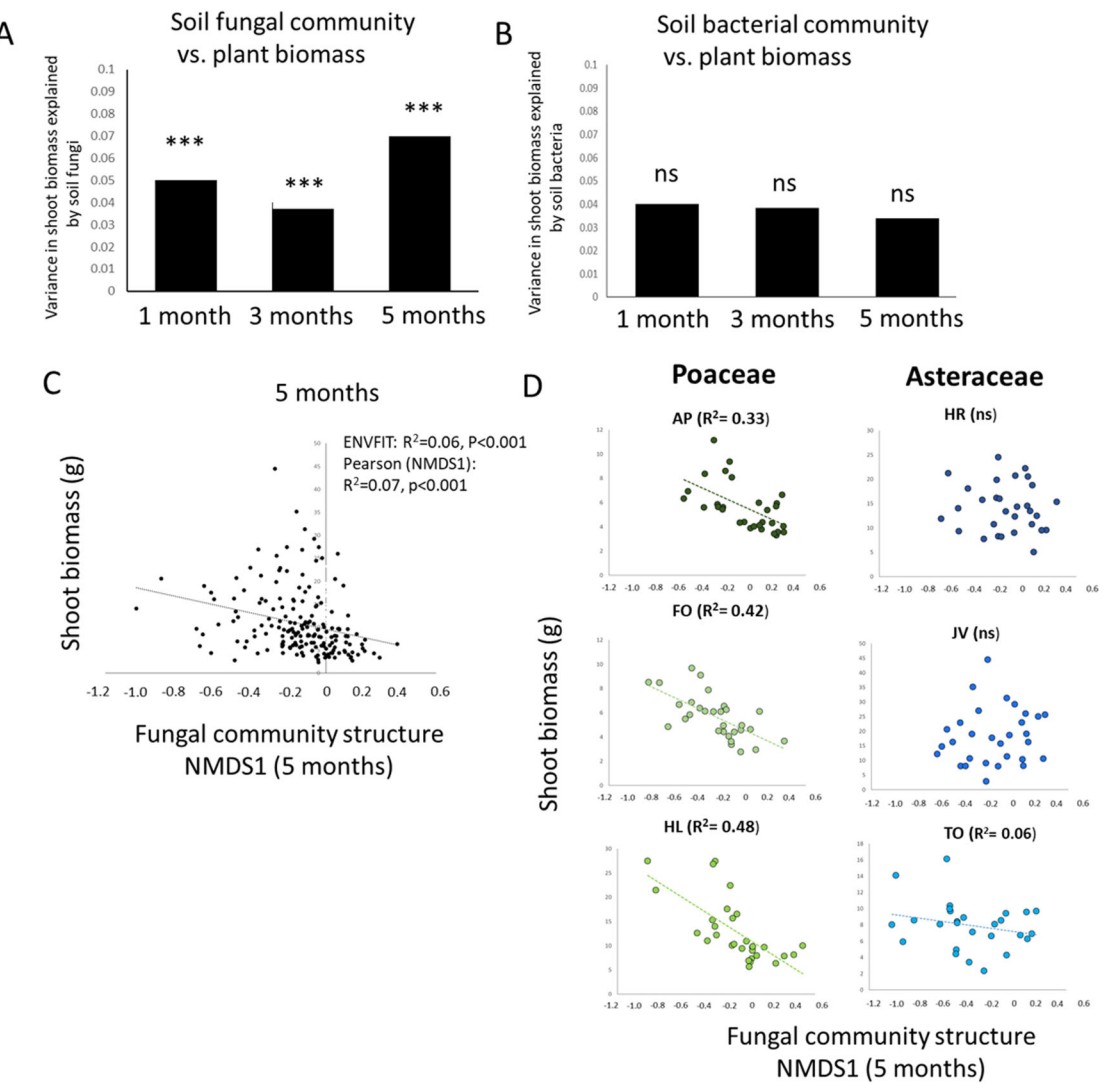

**Fig. 5 The soil microbial community explaining plant growth.** The variance in plant shoot biomass (dw) explained by soil fungal (**A**) and bacterial (**B**) community composition at different time points. The statistical results given in panels **A** and **B** are PERMANOVAs on Bray-Curtis distances and based on full data. **C** Relationship between shoot biomass and fungal community structure measured by first dimension of community structure NMDS1 (Pearson correlation) and by using all the data and full model (ENVFIT; see text), and **D** divided per current plant species. In **D** correlation coefficients are shown only for plant species significantly responding to changes in the soil fungal community.

but not if Asteraceae grew in their own soil (i.e., a family-specific soil effect; Fig. 7D).

There were fewer *Actinobacteria* inside the roots of all plants when grown in conspecific compared to heterospecific soils (LME: $F = 4.01$, $p = 0.047$, Fig. 7E). Especially the relative abundance of *Streptomycetes* (own soil effect across species LME: $F = 5.95$, $p = 0.016$) and *Pseudonocardiaceae* (own soil effect across species LME: $F = 2.46$, $p = 0.046$; Supplementary Fig. 12) decreased when growing in own soil compared to growing in other soils across plant species. We detected family-specific own-soil effects for endophytic *Acidobacteria* (interaction own-soil*plant family: LME: $F = 7.55$, $p = 0.007$) and *Chloroflexi* (interaction own-soil*plant family LME: $F = 4.47$, $p = 0.038$). *Bradyrhizobium* (LME interaction own-soil*-plant family: $F = 6.89$, $p < 0.01$) and *Acidibacter* (LME interaction own-soil*plant family: $F = 5.77$, $p = 0.017$) were increased if Poaceae were grown in their own soil, while *Pseudomonads* (and especially OTU83) were four times more abundant in the roots of Asteraceae (*J. vulgaris* or *T. officinale*) when grown in their own soil (Dunn's post hoc (home-away) JV: $Z = 2.38$, $p = 0.024$, TO: $Z = 2.27$, $p = 0.046$; Fig. S12).

## Discussion

We show that plant-mediated effects on the soil microbiome are reversible, but also that soil legacies from previous plants at species and plant family level can be detected in the soil fungal community for at least 5 months after removal of the previous plant and subsequent colonization of the same soil by different plants. The effect of the previous plant on soil fungal community structure appears to be larger than the effect of the current plant species. This is important, as soils are dynamic and every soil arguably has a pre-existing legacy caused by previous plants or plant communities. Our results indicate that when a plant arrives or is planted into the soil, even months after growing in this soil, it may still experience the microbial legacy effects created by the plants that grew previously in that soil. We can only speculate how long it will take before the legacy of the previous plant in the soil fungal community has disappeared entirely, as the 5 months of 'current' plant growth following one year of 'previous' plant conditioning was not enough for these soil legacy effects to fade away. This finding has important consequences for plant-growth experiments using field collected soils with previous legacies, but

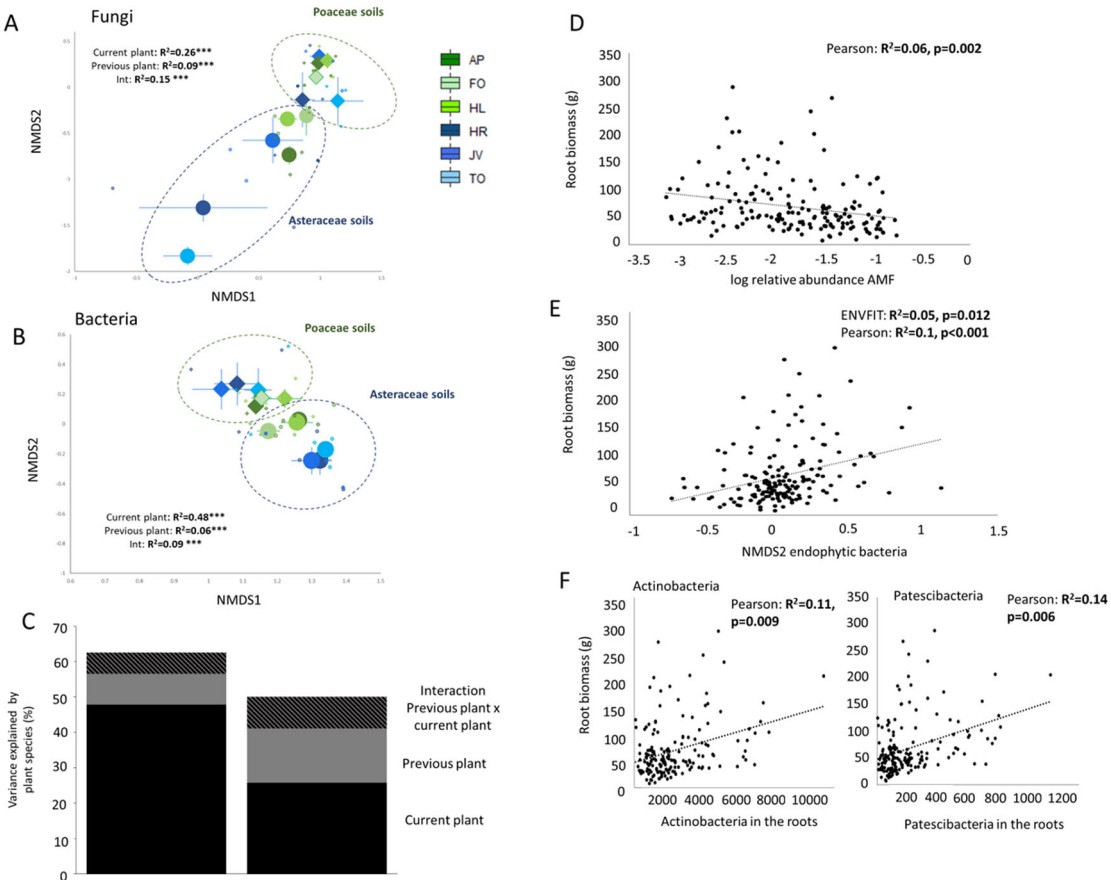

**Fig. 6 Endophytic microbes and their effects on plant growth.** Fungal and bacterial root endophytes affected by previous and current plant species and functional groups and their relationship with responding plant growth. **A**, **B** NMDS using Bray-Curtis distance on the effect of current plant (colors: green shades depict Poaceae, blue shades depict Asteraceae) and previous plant functional group (shapes; circles represent Asteraceae soils and triangles Poaceae soils) on fungal (**A**) and bacterial (**B**) community structure. PERMANOVA results ($R^2$ and significance) for the model are given in the figure and *** notes significance at the level $p < 0.001$. **C** variance in root endophytic bacterial and fungal community structures explained by conditioning (previous) and responding (current) plant and their interaction. **D** the relationship (Pearson correlation) between (log relative abundance of) AMF inside the roots and plant belowground biomass. **E** relationship between root biomass and bacterial community structure measured by NMDS2 (Pearson correlation) and by using all the data and full model (ENVFIT), and **F** read numbers of two bacterial phyla (Actinobacteria and Patescibacteria) most strongly correlated with belowground biomass (after FDR correction). For all of the bacterial phyla and fungal classes see fig. S5. The 2D stress values for the root microbiomes were 0.12 for fungi and 0.08 for bacteria.

also for understanding plant community dynamics in natural and anthropogenic ecosystems, and suggests that the legacy of previous plants or plant communities on the soil microbiome lasts for months after new plants colonize the soil, something that has been previously overlooked.

For soil fungi, the effects of the previous plant on the soil community outweighed the effects of the current plant, while for soil bacteria, each plant quickly modified its own rhizosphere microbiome although the influence of the previous plant was still detectable. These findings are in line with the conceptual idea that fungal growth rates are slower than those of bacteria[36,37] and that because of this, fungi are more stable and less affected by, for instance, temporal variability in the habitat or environment[33]. Importantly, the more persistent effects of plants on the soil fungal communities than on bacterial communities may explain why most correlative studies that link plant responses and changes in the soil microbiome have shown that fungal communities drive plant community dynamics, while soil bacteria do not seem to strongly influence plant-soil-feedbacks[7,8] despite their known importance in rhizosphere processes[1,9]. This is potentially due to faster turnover times of soil bacterial communities[36]. Furthermore, due to their hyphal growth form,

many fungi can simultaneously grow inside the roots and in the rhizosphere environment[38], and soils could therefore potentially predict intimate active fungal-plant relationships better than bacterial-plant relationships. Another possible option is that not all organisms detected with DNA-based methods are active and thus part of the signal we are detecting originates from dead cells or inactive organisms[39]. Alternatively, bacterial DNA could potentially be recycled quicker than fungal DNA which is protected within the hyphae, but this needs further testing. However, recently it has been shown that 80% of fungi detected in the rhizosphere in a similar grassland system were actively participating in recycling plant-derived carbon[40].

Interestingly, the effects of the previous and current plant, especially on fungi, were conserved between the two plant families. Here, we show that even after 5 months, the fungal communities in soils in which currently Poaceae grow with a legacy of previous Poaceae growth differ from the fungal communities in soils in which Poaceae grow but with a legacy of Asteraceae. Similarly, 'current Asteraceae' soils with a legacy of Poaceae had very different fungal communities than 'current Asteraceae' soils with a legacy of Asteraceae. The differences between the two plant families, which represent two distinct

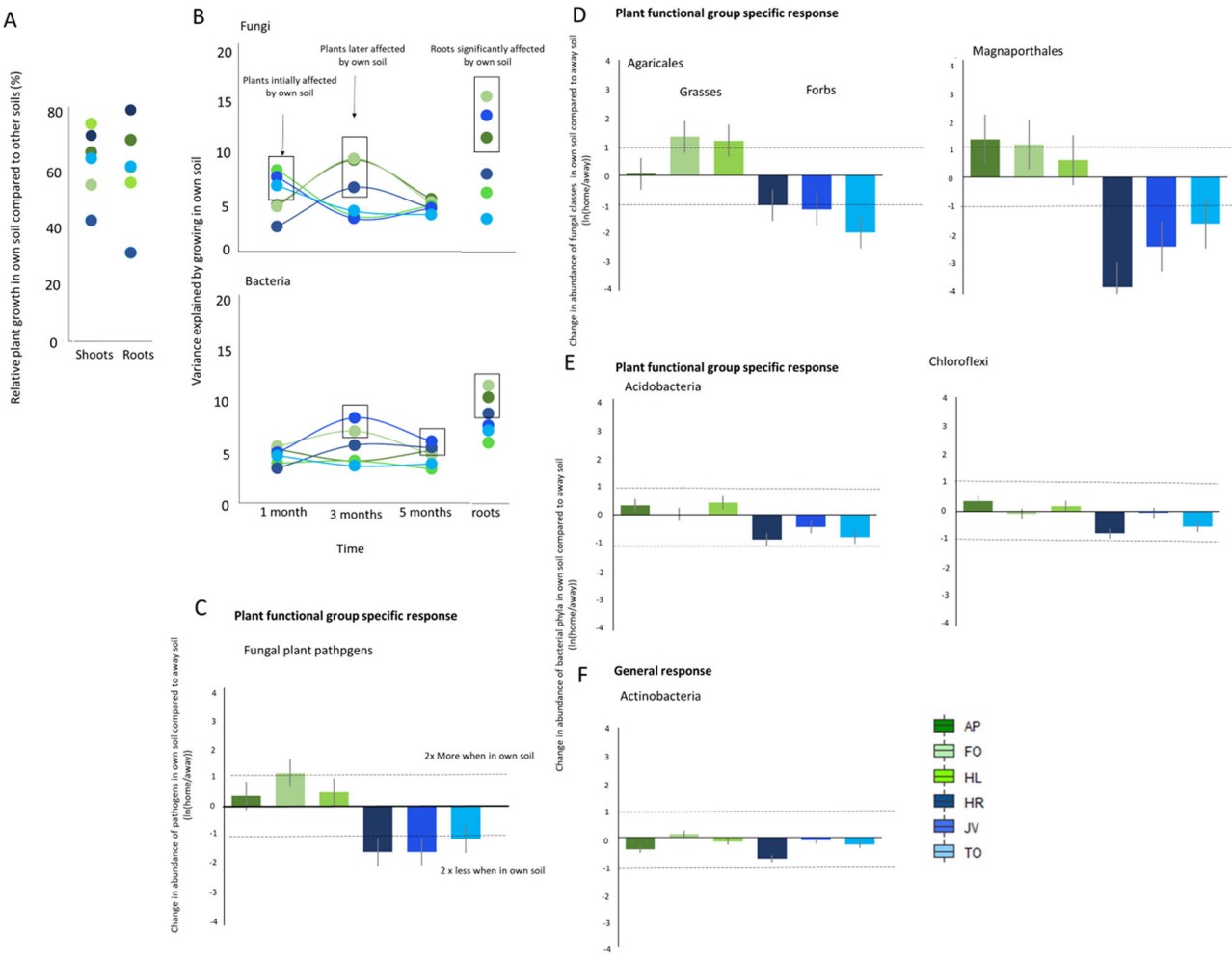

**Fig. 7 Home-away effects on plants and microbial groups.** Plant growth in their own vs other soils. **A** the relative plant growth of the plant in its own soil compared to that in other soils for shoots and roots. **B** the variance (measured as $R^2$ values) explained by growing in own soil (vs other soils) over time and between plants species estimated with PERMANOVA. Significant effects ($p < 0.05$) are indicated in black boxes. **C** functional groups of fungi, **D** fungal orders, and **E** bacterial phyla affected significantly by plants growing in the soil where plants from same family had grown earlier calculated using a log-ratio [ln(home/away)]. **F** The only bacterial phyla significantly affected consistently across plant species when the plants were grown in their own soil calculated using a log-ratio [ln(home/away)]. Bars represent mean home-away effects and error-bars represent standard errors calculated based on individual replicated blocks ($n = 5$). Plant species specific effects of growing in their own soils on fungal classes and bacterial phyla are shown in Supplementary Fig. 12. The colors refer to plant species, green colors mark Poaceae and blue colors Asteraceae. The effect of abundance in own soil vs other soils is calculated by dividing the taxa abundance in own soil by the abundance in all other soils. Dotted lines mark when there are two times more or two times less members of fungi or bacteria in the soils.

growth forms confirm and build on previous findings on the role of plant functional groups in plant-soil feedbacks. For instance, meta-analyses show that the dichotomy between grasses and forbs generally creates robust soil legacy effects e.g. [41]. Furthermore, other recent work has shown that plant family and functional group can explain a large portion of the variation in fungal community structure[33,42,43]. Functional traits as well as growth and nutrient acquisition strategies[44–46] and chemical defenses[47,48] of the selected plant species likely have more similarities within than between groups. This may explain why the two groups have a driving effect on plant-soil feedbacks in plant communities, in greenhouses, and field studies[7,35].

An important outcome of our study is that soil legacies are taken up in early life stages of the plant[49] and remain present inside the roots of the plants when they grow in the soil and by this change the soil microbiome. This is supported by the observation that both the bacterial and the fungal endophyte community sampled after 5 months of plant growth reflect the legacy of the previous plant. Hence, besides the seed-microbiome the plant inherits from its parental plant[19], also the legacies in the soils where it lands shape its endophytic microbiome. This is important, as endophytic bacterial communities are generally more tightly linked to plant performance than soil and rhizosphere bacterial communities[26], while for fungi both rhizosphere soil and endophytic communities influence plant performance to a similar extent due to their ability to bridge endophytic and soil environments[24,50,51]. Here we confirm that the composition of the endophytic microbes is specific to the plant species carrying them partly due to inherited seed microbiome[19,27,52] but also show that endosphere microbiome of the current plant depends on the previous plant that grew in the soil.

Bacterial endophytes, and especially *Actinobacteria* and *Patescibacteria*, were modulated by the legacy of the previous plant, and influenced current plant growth. The role of *Patescibacteria* in plant health is still unclear, but they have been recently found inside the tissues of different plants[29,53]. However,

*Actinobacteria*, and especially *Streptomycetes*, are often detected inside plant roots and can act both as pathogens and can be beneficial to the plant[54,55]. Here, we show that root biomass increased across plant species when the relative abundance of *Actinobacteria* inside the roots increased suggesting a generally positive role of these microbes in influencing plant growth. Interestingly, we also show that all plant species had fewer *Actinobacteria*, and especially *Streptomycetes* inside their roots when grown in soils with a legacy of their own species. Endophytic *Streptomycetes* have been found to be strongly selected by their host plant, they play a role in the magnitude of plant-soil feedbacks and help the plant-host cope with drought[25]. We speculate that plant species-specific selection on soil microbes, such as *Streptomycetes*, may provide an interesting avenue for further investigation on the role of these endophytic bacteria on conspecific plant-soil feedbacks.

We observed a trade-off between the relative abundance of AMF inside plant roots and the biomass of the respective root system, but not shoot biomass. Plants with higher relative abundance of AMF in their roots had a lower root biomass, probably due to a decreased necessity to scavenge for nutrients and especially phosphorus in the presence of AMF[56]. We detected a reduction in the relative abundance of AMF when growing in their own soil only for one plant species and one time point and the previous plant identity or family only had a minor effect on the community structure of AMF within the roots. As we measured relative abundance we cannot draw conclusions from these results on the role in plant-soil feedbacks. Other studies have shown that AMF explain plant-soil feedbacks, through mutualistic relationships[13,57], and that plant soil feedbacks distinctly differ between plants that form arbuscular mycorrhizal interactions and plants with other microbially mediated nutrient acquisition strategies[58]. The species that were used in this study, generally have negative conspecific plant-soil feedbacks[15] and also all form arbuscular mycorrhizae; they all are equally benefitting from association with AMF and the identity of the fungal partners seems not to be important. Alternatively, it is possibly that a potential links between AMF and plant-soil feedbacks, can be detected when focusing on AMF colonization rather than focusing on community composition as was done in the current study. While we here focused on changes in community structure, further studies should also measure absolute abundances and activity of other microbes.

All six plant species exhibited strong negative conspecific feedbacks. The variance in microbes explained by growing in conspecific soils was highest at 3 months of growth, and declined substantially after, highlighting a temporal dimension in plant-soil feedback effects, which we speculate to be due to dynamics in the soil, or decreased plant sensitivity with older age[18]. We observed only in Poaceae that feedbacks were due to accumulation of potential plant pathogens in the soil and inside roots when they were grown in soils with a legacy of their own species. The soil fungal community as a whole, and specific plant pathogenic fungi, have recently been shown to modulate plant community dynamics in grasslands[7,8,59]. Here we show that Poaceae increase the relative abundance of plant pathogenic fungi in the soil when grown in monocultures. More importantly, we show that different Poaceae species accumulate specific fungal pathogens in their roots that can, in turn, cause negative plant-soil feedbacks when these Poaceae are grown in conspecific soils. In all six soil legacies, the soil fungal communities differed when a plant was grown in its own soil, from the communities formed when grown in a soil with a legacy of another plant, and when this effect was largest, depended on plant species. This is in line with work on invasive plant species and their endophytes showing species-specific effects and acquisition of endophytes[30].

Some of the plants in this experiment showed a strong relationship with soil microbial communities at the onset of the experiment, while for others the growth was related more strongly to the current microbiome in the soils, which still contained a detectable legacy of the previous plant's soil microbiome. The variability in the magnitude of plant-soil feedbacks between plant life stages has been noted earlier[16,18] and here we offer a microbial background to this phenomenon. A potential caveat of our study is that we did not measure soil chemistry-mediated legacy effects, which have been shown to play a role mediating plant legacy effects in some studies[13]. However, our recent work in similar systems and conditions revealed no significant role of soil chemistry in driving legacy effects in plant communities[7]. Therefore, in the current study we focused on relationships between microbial composition of soil and root endophytic microbiomes and plant growth. Yet, we acknowledge that plant-soil feedbacks are driven by both soil abiotic and biotic factors, and that these often act in synchrony[10,11].

On the basis of our results, we propose a rethinking of how soil microbiome-mediated legacy effects work. First, no soil with plants growing in it is naive and without a legacy. Therefore, in order to evaluate what are the main factors predicting plant and especially crop growth, we need to look into the history of the soil and importantly also evaluate the plant 'holobiome'[26,60]. We show that part of the microbial legacy effect is plant-species specific while another part of this effect is plant family-specific. Especially, Poaceae generally have negative effects on other Poaceae growing in the same soils while the effects on Asteraceae are mainly plant species-specific. In a wider perspective, our results show that soil and root microbiomes are important for plant growth and that plants can be used to directionally change the microbiomes and hence steer plant growth.

In conclusion, our study shows that soil legacies wrought by previous plants can remain present in the soil for months, even when subsequent plants colonize (and condition) the soil. Bacterial communities change quicker than fungal communities and our findings suggest that plants take up microbes and especially bacteria from these pre-existing soil legacies in their endophytic compartments at a very early seedling stage only, and that these endophytes may play a more prominent role in driving plant performance than the microbiome present around the roots of the older plants. Our study also highlights microbial taxa that consistently drive negative conspecific plant-soil feedbacks across plant-species, and characterizes the role of these species in plant-soil feedbacks thereby providing an exciting venue for further research.

## Methods

**Experimental design.** The set-up of the conditioning phase is shown in Fig. 1. Thirty containers (48 cm × 80 cm × 50 cm) were filled with soil that was sieved through a 32 mm sieve. The soil was sourced from a grassland near Lange Dreef, Driebergen, The Netherlands (52° 02′ N, 5° 16′ E) and is characterized as holt-podzol, sandy loam (84% sand, 11% silt, 2% clay, ~3% organic matter, 5.9 pH, 1,151.3 mg/kg total N, 2.7 mg/kg total P, 91.0 mg/kg total K; analyses by Eurofins Analytico Milieu B.V., Barneveld, The Netherlands, using in-house methods[61];). Monocultures of ~100 individuals of six common plant species all forming symbiosis with arbuscular mycorrhizal fungi (AMF) representing Asteraceae (forbs) and Poaceae (grasses) plant families (Fig. 1) were grown in these soils for one year (from May 2017 to May 2018;[33]). Each species was planted in a separate container in 5 replicate blocks in a randomized block design. We used three Poaceae species (*Holcus lanatus* (HL), *Festuca ovina* (FO), *Alopecurus pratensis* (AP)) and three Asteraceae species (*Hypochaeris radicata* (HR), *Jacobaea vulgaris* (JV), and *Taraxacum officinale* (TO)), all of which are very abundant and commonly occur in grasslands in the Netherlands. The experiment was conducted in the common garden at the Netherlands Institute of Ecology (NIOO-KNAW, Wageningen, The Netherlands, 51° 59′ N, 5° 40′ E). Mesocosms were watered regularly during the summer months to avoid desiccation.

The results from the first phase of the experiment are presented in an earlier publication[33]. There we showed that at the end of the conditioning phase both bacterial and fungal community structure was affected by plant species and plant

family while neither fungal nor bacterial abundance measured by pPCR was affected by plant species or plant family. Highest abundance of bacteria in the end of the conditioning phase was 2062 copies per μl DNA for AP and lowest was 1779 copies per ul DNA for JV and highest fungal abundance was 47.72 copies per μl DNA for HR and lowest 7.52 copies per μl DNA for HL.

After one year (June 2018) the aboveground 'previous' plant parts were removed, and the soils were divided into six equal 'soil monoliths' placed in smaller containers (Fig. 1 and Supplementary Movie 1). Four seedlings of each of the six species were planted in each new container so that there were six monocultures in each larger container (180 units; Fig. 1). Roots were not removed from the system to include the legacy effect of decomposing belowground biomass, as would be the case under natural conditions of plant removal (e.g. strong grazing or wild boar disturbance). Three weeks later, four seedlings of a single species grown from sterilized seeds on sterile glass beads were planted in each section. This equals to 180 experimental units (6 sections within 6 monocultures arranged in 5 blocks). Soil samples were collected using a small soil corer (12 cm deep, 7 mm diameter) from 5 locations in each section (one next to each plant and one in the middle), pooled, homogenized, and immediately stored at −20 °C until molecular analysis. Samples were collected just prior to planting with the previous plants still in it (30 samples; May 2018)[33], 1 month after planting (July 2018), after 3 months (September 2018), and after 5 months of plant growth (November 2018). The experiment was harvested in November 2018 and total oven-dried aboveground and belowground biomass was determined for each mesocosm. Samples from roots were collected from randomly selected washed root fragments, surface sterilized using published protocols[22,29] and stored in −20 °C prior to molecular analysis.

**Sample preparation and sequencing**. DNA from soil was extracted from 0.75 g of soil using the PowerSoil DNA Isolation Kit (Qiagen, Hilden, Germany) and from 0.5 g of homogenized roots using MP Biomedical FastDNA™ Spin Kit following the manufacturer's protocol[29]. Fungal and bacterial DNA was amplified using primer sets ITS4ngs and ITS3mix targeting ITS2 region of fungi[62] and primers 515FB and 806RB targeting V4 rRNA region of bacteria[63–65], purified using Agencourt AMPure XP magnetic beads (Beckman Coulter). The sequences of used primers are presented in supplementary table 1. For bacterial PCRs targeting root endophytes, blocker sequence was used to prevent amplification of chloroplast and mitochondria DNA[22]. Adapters and barcodes were added to samples using Nextera XT DNA library preparation kit sets A-C (Illumina, San Diego, CA, USA). Separate equimolarly pooled libraries were constructed for bacteria and fungi. Bacterial samples ($n = 720$) were analyzed in 5 MiSeq runs (4 for bacteria in soils and 1 for bacteria in roots) and fungi ($n = 720$) in 3 MiSeq runs. Libraries were sequenced using MiSeq PE250 at McGill University and Genome Quebec Innovation Center. Extraction negatives and a mock community, containing 10 fungal species, were included in each library and used to compare data between sequencing runs, detect possible contaminants and to investigate the accuracy of the bioinformatics analysis.

**Bioinformatics and statistics**. Bacterial and fungal sequences were analyzed using the DADA2 (v. 1.12;)[66] using SILVA (v.132) as reference database and PIPITS (v. 2.3;)[67] with UNITE (v. 8.0;)[68] as reference database, respectively. Finally, the fungal OTUs were parsed against the FunGuild (v1.1) database to assign putative life strategies[69] and this was further curated using in-house databases[51]. All singletons and all reads from other than bacterial or fungal origin (i.e. plant material, mitochondria, chloroplasts, and protists) were removed from the datasets. To account for large differences in read numbers, all samples with less than1000 reads or more than 60,000 reads were removed which resulted in removal of 6 samples for bacteria and 24 samples for fungi. Furthermore, all OTUs and ASVs present in <5 samples with relative abundance of <0.001% were removed from the dataset. In the end, we detected total of 2525 ASVs for bacteria and 2613 phylotypes of fungi across samples. Cumulative sum scaling (CSS) was used to normalize the data[70,71]. PERMANOVA model was constructed to investigate the effects of current and previous plant and the family of the current and previous plant on soil microbial community structure using Bray-Curtis distance in 'vegan'[72]. A full model was run with all the time points to detect the effect of time with block as a strata and treatments nested in time but to answer main questions here, we also constructed simpler models with all the time points measured separately. To estimate the effect of growing in their own soil, also a PERMANOVA model was used with interaction own soil × plant identity/plant family as the main investigated factor. To investigate the sensitivity of individual plant species to microbiomes, also plant species specific PERMANOVAs were run for all time points. For all models run, betadisper was used to check the homogeneity of dispersion. We use the $R^2$ values from the models used to estimate the amount of variation explained by a variable in the model. To visualize the effects of previous and current plant on microbial community structure, NMDS ordination (without further transformations) was used. To simplify the message, we chose to use for figures either NMDS1 or NMDS2 but always run the full model for statistical significance. To estimate the effect of family of previous and current plant on NMDS1 of bacteria and fungi lme model with both species of current and previous plant as random factors was used. ENVFIT with 999 permutations and block as strata was used to fit plant (shoot and root) biomass data (all at the 5-months-stage) as vectors to the community structure data in 'vegan'[72]. Relative abundances of bacterial phyla and fungal

classes and orders were calculated. Furthermore, relative contributions of fungal functional guilds were calculated. All groups of bacteria and fungi present in <10 samples were removed from the analysis.

We tested the effect of previous plant and current plant and the plant family on plant biomass and relative abundances of bacterial and fungal taxa using a linear mixed effect model (package NLME) with block as a random factor. When plant family effects were investigated, we further added the identity of both current and previous plant as random factors in the model. We did the testing independently for all time points and for root endophytes for the relative abundances. For all analysis we used posthoc Tukey HSD tests to test which treatments differed from each other. We explored for a normal distribution of residuals using QQ-plots and a Shapiro–Wilk test and homogeneity of variances using a Levene's test. For the relative abundances of bacterial and fungal taxa, arcsin square root or ln transformations were used. The transformation of the data is mentioned in the text and in the figures. All $p$-values derived from multiple calculations (such as in Fig. 3D) were corrected with Benjamini and Hochenberg which relies on calculating the expected proportion of false discoveries among rejected hypotheses to control for false discovery rate (FDR)[73]. For some of the individual OTUs if normality was not achieved, non-parametric tests such as Dunns post hoc was used to investigate the effects of growing in its own soil compared to other soils. In these models the identity of both current plant and previous plant and block were included in the model as random factors (Fig. 1).

To calculate the difference in microbial community structure in own soil compared to away soils a model with 'away' plant species or its functional group as random factor was built and PERMANOVA was used. For the relative abundances and plant growth, linear mixed effect models were used with 'away' plant species as random factor with transformations described above. To calculate 'home-away' soil effects, formula ln(home/away) was used. All statistical test were performed in R (v. 3.6.0).

**Reporting summary**. Further information on research design is available in the Nature Research Reporting Summary linked to this article.

## Data availability
The sequence data generated in this study have been deposited in the ENA database under accession code PRJEB38409. The plant growth data are available DRYAD database https://doi.org/10.5061/dryad.4qrfj6qb1.

## Code availability
The R-Code used in this article is available from authors upon request.

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

## Acknowledgements

We thank Grace Wangsa Putra, Eefje Sanders, Anna Kielak, and Stijn Hofhuis for help in harvesting the experiment and/or molecular work. The sequencing was done in collaboration with McGill University and Génome Québec Innovation Centre. This study was funded by The Netherlands Organisation for Scientific Research (NWO VICI grant 865.14.006). S.E.H. was funded partly by Maj & Tor Nessling foundation.

## Author contributions

S.E.H., R.H., M.H., K.S., J.R.dL., and T.M.B. conceived the initial idea of the experiment, S.E.H., R.H., M.H., K.S., J.R.dL., and R.J. set-up and harvested the experiment. S.E.H., K.S., and R.J. performed the molecular work and S.E.H. analyzed the data. S.E.H., R.H., and T.M.B. wrote the initial draft of the manuscript and all authors commented on it.

## Competing interests

The authors declare no competing interests.
