## [Peer Review File · Nature Communications]

Reviewer comments, first round -

Reviewer #1 (Remarks to the Author):

The manuscript " Persistence of plant-mediated microbial soil legacy effects on soil and inside roots" by Hannula et al reports that legacies of previous plants were detectable in soil fungal communities, bacterial legacies was short-term in soils but both fungal and bacterial legacies were consistent inside the root endophytes. They concluded that plant-soil feedbacks are long-term, but mediated by an endophytic community recruited at early stage of the plant development.

This is well designed and executed research based on strong conceptual framework. Two of issues- I would like to raise-

1. Soil microbial biomass or abundance can also play important role in plant productivity. This is not accounted for by this study. I wonder, inclusion of abundance data can further support the conclusions? If this is not possible, I suggest this needs to be discussed and acknowledged in the discussion section.

2. There are too many hypotheses and messages in this manuscript which hampers the flow of the manuscript in my opinion. I was wondering if authors can crystallize 2-3 key/ novel hypotheses and construct the story around those to improve the flow of the message?

Other comments

L-116-177. Was mitochondria blocker was also used to reduce the number of plant sequences in bacterial data? If yes, please provide details.

L215-17. R2 values are pretty low- I suggest authors mention this in text- something like "the legacy o the previous plant also had small but significant effect on the composition---"

L382-383. it will be good to discuss a bit more that why AMF had a minor role in plant-soil feedbacks?

Figure 3, font size of X-axis for all graph is too small to read

t

Reviewer #2 (Remarks to the Author):

This manuscript by Hannula et al. presents the results of a large mesocosm experiment designed to examine the effects of individual plant species on soil microbial communities and the plant microbiome. A unique feature of the study is that the authors attempt to dissect the time course of soil legacy effects on bacterial and fungal communities by testing the effects of previous plant species on the above variables simultaneously with the current plant residents. While many previous studies on plant-soil feedbacks have been conducted, this stands out by the detailed examination of the timing of legacy effects (line 300-303), and the differences between bacterial and fungal (including AMF) behavior and effects are also relatively novel. The prevalence of negative feedback effects for all species (388) is interesting and adds to previously published results. In total, there are a lot of interesting data.

It is interesting that soil legacies might have the greatest effect on seedlings and during seedling establishment. Also it is interesting that there still might be a legacy effect exhibited in root microbiomes even when it has disappeared in the soils. When seedlings were planted in phase 2, could they have brought microbes with them even following surface sterilization?

There are so many comparisons and contrasts that sometimes it is difficult to follow all of the

parts. Some type of graphic presentation of all the independent and dependent variables in relation to time and experimental design, and bacteria vs. fungi, would help clarify this point. There are a lot of graphics already, but perhaps it could be incorporated into Fig. 1.

A couple of questions came up in my mind. What about Oomycetes, and what about nematodes? They have been shown to be extremely important in other PSF studies.

Also, monocultures rarely exist in nature except in highly managed agricultural situations so some discussion should be devoted to the more realistic scenario where multiple co-occurring plant species were present in the previous season and again in the current season? Also, the physical effects of soil are given less attention than biological effects. The spread of roots in the soil (depth especially) could also be an important factor where grasses tend to have shallower, fibrous root systems while some forbs have a deeper taproot. Admittedly, there is a lot of experimental power in the approach used.

Was soil sampled for microbial testing after the first phase of growth with the 6 monocultures? See line 338, and Fig. 2. It seems that this is a black box (see Fig. 3, yellow bar). If there was no "starting point" quantification, some discussion of why is warranted. Also, why weren't roots sampled over a time course instead of one time at end (L106). What are the origins of the soil used? Field soil? Was it sterilized before use? Also, say something about plants being AMF or not earlier.

No detailed data are presented on the microbiome results and the discussion of individual taxa (265-286, 358-375) seems like it is cherry-picking. Fig. 6 F as well. Additional information is presented in the Supplement and potentially some of it warrants presentation in the main text.

The design seems fundamentally nested where previous plants grouped by family and then species within family, but I am not sure that is how the statistics were done.

Specific comments

Part B and C in Fig. 3 are hard to follow.

L109-137 – sequencing and bioinformatics seem solid but not a lot of detail. Any references of past results to cite as verification?

L150 – remind us of what the hypothesis is?

L175 –not clear to me where three microbiomes comes from given two types cited later in sentence?

Fig. 4 X axis hard to read – too small, which makes it very hard to interpret the data. Same comment for Fig. 6. And Fig. 7. Axes values need to be bigger and more legible. Even at 300% they were hard to see. It should be made clearer that the left column in B are grasses and the right are forbs.

In Fig. 3 and 5, NMDS1 needs to be better explained.

Fig. 7B is interesting but why the decline from 3 to 5 months?

Reviewer #3 (Remarks to the Author):

The study design is neat and addresses an interesting aspect of plant microbial interactions in soil. Thus, the study has great potential but I cannot recommend that it is published in its current form. I must admit that I'm rather put off by the bold setting of the intro and a vagueness of statements that makes it difficult to assess how thoroughly the authors understand the broad topic addressed. For instance, the molecular methods applied are not usually very successful in detecting AM fungi and without discussing in any detail what level of detecting (nr of OTUs and their relative abundance) that is detected the authors go into interpretations of this group. This makes me very skeptical and I would like more convening presentation in the main text. I do not have the expertise to evaluate the statistical analysis and I think the presentation of methods and results need to be more convincing at least for fungal biologist as myself. I have provided many details

below that I hope the authors will find useful when revising the manuscript. Discussion needs to be streamlined, some arguments return several times and overall, it gets too speculative.

Comments throughout the text

In the abstract the authors use the word wrought which is correct but I must admit that I did not know this word and had to look it up. Why not use the synonym shaped that is easily understood? On line 14 the authors use the term long-term but they have previously on L 6 they said that this is about months. I do not see how several months can be considered a long-term effect if one think of soil processes or ecosystem processes, I think a more specific term or reference to the considered time span is preferable.

The abstract ends with an interesting claim of being the firsts to actually show that plants shape soil microbiome. I agree that it is a commonly held assumption but maybe based only on indirect evidence because there is a multitude of studies showing that soil microbial communities as shaped by the plant communities (and vice versa). I see a problem with the present statement because it is very unspecific to say soil microbiome, what characteristic in the microbiome is it that is affected and reversible? I would like to see a more specific terminology such as community composition instead of microbiome.

The introduction starts out in a rather generalized tone and lacks specificity. L 26: Why do the authors assume that these plant-mediated effects are restricted to the microbiome and can the effects of the microbiome really be separated from the direct effects of the litter in feed-back. This comes back on L 34 and it appears to me that the authors work under the assumption that feedback is solely a microbial process. Its surely refreshing with this attitude in contrast to many earlier studies that would assume that the feed back effect is all about litter composition and the chemical signature it leaves behind but maybe a more balanced view would be to at least in the introduction acknowledge that feed back is not just a microbial process. L 30: Im don't know of any evidence that plant-soil feedbacks are general among functional groups such as grasses and forbs. Are those really examples of relevant functional groups with regard to plant feed back. I would expect early succession vs late succession plants to be more relevant and there are certainly examples of grasses and forbs in both these groups.

L 43-44. Its also very unlikely that an older plant would all of a sudden end up in a soil conditioned by another plant so it seems only logical that the effect of feed-back is stronger on seedlings.

L 51. I would also think that this is a founder effect, while older roots are already colonized there is no open niche to colonize so they are less susceptible to new colonizers.

L58, I can not quite follow the sentence ending in soil origins? Please develop for clarity.

L 81. Im not convinced that selecting one family of forbs is a good rational to represent all forbs. It makes sense for comparison to grasses that is one family but there is no rational in what way Astereaceae represents all forbs. The presentation would be more convincing by being more accurate and less bold. Other than that I like the study design. Its not clear from the presentation in the intro if the entire plant was removed or if old root system is still present in the mesocosm. In natural soils we would expect roots to sit around but maybe decompose quite a bit from one season to the next. This is not clear in the methods either, see L 103 where it says removed.

L 90. Why did the authors hypothesis that its pathogens in the rhizosphere not in the roots that will affect plant growth?

L95, its very unspecific again to say that bacteria and fungi are different in growth strategies / traits. Fungi encompass both plant symbionts and plant pathogens to a much larger extent than bacteria and while there are some important plant pathogens among bacteria they are few compared to fungi.

Experimental design, its not clear from the text how many replicates the meso cosm was set up in.

L 106 with the current sampling of roots do you assume that you sample the endosphere community or root associated community. It would be nice if this could be clarified.

L110 no empty line after heading level 2, or there should be an empty line after L 100 etc. Be

consistent though out the text

L 121 no line break before this line

L128, what confidence level was applied to accept an assigned taxonomy?

L131 what kind of datasets are the authors referring to, sequences of known species? Unpublished information of life strategies?

L134, what component of the microbial community, all time points of soil community or only root community? Please be more specific

L 135, here the authors use plant family rather than grass vs forbs. This is more accurate and should be used through out the text. Compare to L 147 where the hypothesis is referred to as being set up to test the difference between plant families.

L 152. What is the effect of root extension on how much of the soil is affected by the new plant. At what point in the experiment is the entire soil actually influenced by the plant? How densely planted are the systems and how much of the soil is affected by rhizodeposition. Also I cannot help to wonder what the effect of spatial variation is within each mesocosm. Not clear from the method description who sampling was designed to account for small scale spatial variation.

L 158-159 The spatial distribution is expected to vary between single celled bacteria and filamentous fungi, can this explain part of the difference in how community structure with new plant?

L 161-162 this sentence makes no sense, previous and current had effect, just above it appeared to me that the effect was strongest from previous plant.

L 176 what does it mean to diverge and become similar?

Compare L 168 and L 178, should there be an indent or not?

L198 this does not quite make sense since 5 month is the only timepoint when there is biomass data so what other time point is there to compare link between biomass and soil microbial community. Consider re-phrasing to make clear what is being compared.

L231 given the molecular markers selected I'm surprised that the study identifies any AM fungi in the samples, the ITS2 regions is well known to work very poorly for these fungi and it would strengthen the statement if it included information on how many taxa were identified or how much of the root endophyte community belonged to AM fungi. Anyone working on AM is likely to be in doubt when reading this sentence. This links to L 250 where I would like to see a nr from x to y %. We know that different AM species may affect different plant species differently so I'm not convinced that a lower level of AMF is biologically important for this biomass in these experimental conditions.

L252 only states an effect but not direction, this is not very useful for the reader. Same thing again on L 255

L265, the text has stated that grasses growing in own soil has more pathogens and then the order Agaricales is mentioned. It makes it sound like this is one of the examples of a pathogen, but surely the authors can not have meant this. Any taxonomy assignment above genus level has little relevance to assignment of functional group and Agaricales is a widely species rich group including many forest pathogens, I really hope this is now where they potential pathogens were identified. Again the main text needs more detail to avoid misunderstandings like this. Maybe just omitting the information on orders being more common. When is order a relevant ecological grouping anyway?

L 272 observe that negative feedback loops are well known among AM fungi and their hosts. See extensive work by Bever cited in this paper (11, 12)

L 303 again I have an objection to using long term in the context of months.

L307, both references are to 37. I'm not familiar with the conceptual idea that fungal growth rates are lower than bacterial, bacteria may have shorter reproduction but fungi likely forms larger biomass in soil, thus having larger spatial effects. I think it's also relevant to take into account how species rank curves look for fungi and bacteria. Fungal communities are usually dominated by a few dominant taxa, their effect thus may be larger than more even bacterial communities (if that's what bacterial communities look like).

L316 I'm thinking you could test this in your data by identifying what proportion of the community that is shared between roots and soil at harvest and the life strategies of those colonizing both roots and soil.

L 332-333 taxonomic rank does not necessarily reflect phylogenetic distance these arguments are not valid unless you can also reference work comparing phylogeny and taxonomy for different plant families.

L 343. I think this needs to be acknowledged already in the intro not to lose interest of the critical reader.

L355, why was seed microbiome not examined if authors thought it might be very important. I think there are many references that would give good support for ruling out many of the observed root endophytes as being seed born.

L376 I have not seen any information about AMF colonization. Detecting AMF reads is not the same as colonization which is inspected by staining of roots, if you had stained the roots that would be cool and I think it should be presented more clearly in the results.

L377 I think it's a sloppy style of writing to say: Plants with relatively fewer AMF ... is it OTUs assigned to AMF or reads representing these AMF. It is not fewer AMF its detection of reads. I would also be very cautious about interpreting relative abundance of AMF since they are notoriously difficult to detect in metabarcoding studies. Unless you can state specifically that you have a high and consistent detection you should be very careful.

L379 on the contrary I would say that AMF species have been shown to have negative feedback and this is widely thought to be a reason for why AMF dominated plant communities are species rich.

L394 I agree that you detect specific fungal pathogens in grass roots but you do not actually show that they in turn cause negative plant-soil feedback.

L400 what exactly do you mean with plants show strong relationship?

Rebuttal to the reviewer comments on manuscript:

Persistence of plant-mediated microbial soil legacy effects in soil and inside roots

Thank you very much for sending us the detailed reviewers' comments and for allowing us to submit a revised version of our manuscript. We found the reviewers comments reconstructive and useful in improving our manuscript.

This is the letter in response to reviewer comments:

In general, we have now rewritten our manuscript and provide a point-by-point rebuttal to the reviewer's comments below, with special attention in particular to the comments on biomass measurements of fungi, seed microbiome, clarifications of statistical methods and comments pertaining to molecular markers on AM-fungi. Specifically, we have re-written discussion on lines 386-402 to better reflect our findings and emphasize that we here focused on community structure of fungi and bacteria, and not on specific (yet interesting) individual groups of fungi. In respect to AM-fungi, we have added information on AM-community structure in Fig S10 (E) and few sentences in the results section on this were clarified.

We have also included additional data on microbial copy numbers from the first phase of the experiment as a background information in supplementary material and methods, yet, it was not in the scope of this study to measure microbial biomass for all the 720 samples, as we had further no hypotheses related to changes in microbial biomass in response to plant species. We have added a sentence about this in lines 401-402 and made clear in earlier instances that we are interested here on changes in community structure, functional groups and individual species.

Furthermore, in response to comments by the reviewers we have clarified our hypotheses (and reduced the number to 4 for clarity), improved the quality of figures, added more detailed explanation of statistical models (in text and in Fig. 1), corrected the statistical models to take the species of current and previous plant into account as random factor and made 2 new figures; Fig S6 to show all correlations between plant growth and microbial taxa in response to reviewer #2 and Fig S4 to show similarities between root and soil microbiomes across times in response to reviewer #3. We have further added sentences on seed microbiome and on the AM-status of the plants as requested by the reviewers.

Lastly, we have revised many specific aspects concerning the readability of the manuscript, to which end several of the reviewers' comments have proven to be very helpful. Specifically, we now refer to Poaceae and Asteraceae and 'plant family' throughout (instead of saying grasses and forbs). Lastly, we shortened the abstract considerably.

We have indicated major changes (all except for correcting minor typos) in blue text color in the manuscript file.

On behalf of all co-authors,

Yours sincerely,

S. Emilia Hannula & Robin Heinen

REVIEWER COMMENTS

Reviewer #1 (Remarks to the Author):

The manuscript " Persistence of plant-mediated microbial soil legacy effects on soil and inside roots" by Hannula et al reports that legacies of previous plants were detectable in soil fungal communities, bacterial legacies was short-term in soils but both fungal and bacterial legacies were consistent inside the root endophytes. They concluded that plant-soil feedbacks are long-term, but mediated by an endophytic community recruited at early stage of the plant development.

This is well designed and executed research based on strong conceptual framework.

***Response: We thank the reviewer for the kind words on our conceptual framework, experimental design and execution.

Two of issues- I would like to raise-

1. Soil microbial biomass or abundance can also play important role in plant productivity. This is not accounted for by this study. I wonder, inclusion of abundance data can further support the conclusions? If this is not possible, I suggest this needs to be discussed and acknowledged in the discussion section.

***Response: The reviewer raises an important point that aside from microbial community structure, the biomass of the microbial community (i.e., abundance) can also play a role. We did not explicitly quantify microbial biomass in this specific study. However, we did quantify microbial abundance by means of qPCR for the first phase of the experiment. At t=0 for this study the copy numbers of fungi ranged from 7.52 (HL) to 47.72 (HR) copies per ul DNA and bacteria from 1779 (JV) – 2062 (AP) copies per ul DNA. We have now included this information as background information for the manuscript in supplementary material and methods. The results show that bacterial abundance was significantly affected by time, but not much by plant species while fungal abundance was affected by both plant species and time (Hannula et al. 2019b).

In the present study, we examine variance in microbial composition that is explained by the current and previous plants and the effects on relative abundance of groups of microbes and individual microbial taxa. We did not measure fungal or bacterial biomass at the later time points, as this answers different questions. We now acknowledge and discuss this in lines 401-402.

Hannula, S. E., Kielak, A. M., Steinauer, K., Huberty, M., Jongen, R., Jonathan, R., ... & Bezemer, T. M. (2019b). Time after time: temporal variation in the effects of grass and forb species on soil bacterial and fungal communities. *MBio*, 10(6).

2. There are too many hypotheses and messages in this manuscript which hampers the flow of the manuscript in my opinion. I was wondering if authors can crystallize 2-3 key/ novel hypotheses and construct the story around those to improve the flow of the message?

***Response: We thank the reviewer for pointing this out. Admittedly, we have struggled with this point ourselves while writing up the paper, and had already reduced the number from eight to five in between earlier drafts.

We have now tried to structure the hypotheses more clearly, and limited them to four main novel hypotheses (L87-98) based on our experimental set-up. To remind readers on these hypotheses, we

have structured the results and discussion sections around these questions and come back to them in the text for clarity.

We hope that the reviewer understands this point of view, and we hope that we have now presented an acceptable middle way.

Other comments

L-116-177. Was mitochondria blocker was also used to reduce the number of plant sequences in bacterial data? If yes, please provide details.

***Response: Indeed, both mitochondria and chloroplast blocker were used to prevent from sequencing plant material. This is now mentioned in L128-129.

L215-17. R2 values are pretty low- I suggest authors mention this in text- something like "the legacy of the previous plant also had small but significant effect on the composition---"

***Response: Indeed, the R2 values of endophytic communities explained by plant family are rather low, but significant. We have now mentioned this in the text as suggested, and revised the statements for clarity (L233-234).

L382-383. it will be good to discuss a bit more that why AMF had a minor role in plant-soil feedbacks?

***Response: There are many studies that observe a role for AMF in plant-soil feedbacks (e.g., Klironomos 2002, Bennett & Klironomos, 2017). However, in our system, we rarely find strong correlations between AMF composition, or their relative abundance in terms of relative read numbers, and plant growth responses. First of all, we should mention that plant-soil feedbacks in our system tend to be negative for most plant species. Following the rationale of microbial accumulation, this would suggest a more prominent role for pathogens/antagonists, rather than for beneficials/mutualists (although AMF have been shown to occupy both ends of that dichotomy). Indeed, we have described such pathogenic drivers of plant-soil feedbacks in other studies previously (e.g. Heinen et al. 2020). Secondly, however, sequencing methodology may also not be the best method to measure effects of AMF on plant growth responses per se, as we do not measure direct interactions, but rather investigate community composition and relative abundance of AMF. While we do detect consistently that soils in which forb grow had relatively more AMF after 1 month and later that the relative abundance in their roots is higher, we could not relate this to the identity or functional group of the previous plant. Growing in own soil only negatively affected the relative abundance of AMF in the soils of one species (JV) and one time point (1 month) showing that the effects of AMF relative abundance were not applicable across species and times. Furthermore, the community structure of AMF inside the roots was not affected by growing in own soil and only weakly by the legacy of the previous plant (new addition to Fig S10E). However, future studies should address this further via methods assessing, for instance, root colonization (staining). We have now modified our discussion paragraph on AMF, to discuss their role in our system, and the limitations of our methods more explicitly (L389-400).

Klironomos, J. Feedback with soil biota contributes to plant rarity and invasiveness in communities. *Nature* **417**, 67–70 (2002). <https://doi.org/10.1038/417067a>

Bennett JA, Klironomos J. Mechanisms of plant–soil feedback: interactions among biotic and abiotic drivers. *New Phytol.* 2019;222(1):91–6.

Heinen R, Hannula SE, De Long, J., Huberty M, Jongen R, Kielak AM, et al. Plant community composition steers grassland vegetation via soil legacy effects. *Ecol Lett.* 2020; 23(6): 973-982

Figure 3, font size of X-axis for all graph is too small to read

***Response: We have increased the font size of all figures to improve readability and thank the reviewer for pointing this out.

Reviewer #2 (Remarks to the Author):

This manuscript by Hannula et al. presents the results of a large mesocosm experiment designed to examine the effects of individual plant species on soil microbial communities and the plant microbiome. A unique feature of the study is that the authors attempt to dissect the time course of soil legacy effects on bacterial and fungal communities by testing the effects of previous plant species on the above variables simultaneously with the current plant residents. While many previous studies on plant-soil feedbacks have been conducted, this stands out by the detailed examination of the timing of legacy effects (line 300-303), and the differences between bacterial and fungal (including AMF) behavior and effects are also relatively novel. The prevalence of negative feedback effects for all species (388) is interesting and adds to previously published results. In total, there are a lot of interesting data.

***Response: We thank the reviewer for the positive words about our setup and the recognition of the novelty of our study design.

It is interesting that soil legacies might have the greatest effect on seedlings and during seedling establishment. Also it is interesting that there still might be a legacy effect exhibited in root microbiomes even when it has disappeared in the soils. When seedlings were planted in phase 2, could they have brought microbes with them even following surface sterilization?

***Response: It is possible that the seedlings brought in microbes inherited in the 'seed microbiome' (Nelson 2018), but it is unlikely that this can explain the legacy effects that we observe. The seeds from which the 'current plants' originated were surface-sterilized and seedlings were then grown in sterilized soil, prior to planting them in our mesocosms. Obviously, what has been sterilized, will likely not remain this way for very long. And by transplanting the seedlings we could have transferred microbes. However, the soils in which the seedlings were transplanted was a soil in which other plants had been grown, and hence, full of microbes. We expect that introduction of potential "new" through the seedlings will have a very limited effect on the soil microbiome in such a system. More importantly, the legacy effects of other species that we report in this study cannot be explained by these microbe introductions via seedling transplantation. It is also important to mention that the volume of the seedling soil plugs would be an estimated 4 seedlings x ~20 mL soil = ~80 mL soil, planted in containers with roughly 13000 mL of conditioned soil monoliths.

We have acknowledged the effect of seed microbiome on the root-associated microbiome and this is part of the 'current plant' influence here. We have made this clearer now in the text

Nelson EB. The seed microbiome: Origins, interactions, and impacts. *Plant Soil*. 2018;422(1):7–34.

There are so many comparisons and contrasts that sometimes it is difficult to follow all of the parts. Some type of graphic presentation of all the independent and dependent variables in relation to time and experimental design, and bacteria vs. fungi, would help clarify this point. There are a lot of graphics already, but perhaps it could be incorporated into Fig. 1.

***Response: We like the idea of adding the 'analytical approach' to Figure 1. This has already been done for the types of samples taken, as well as the full design, so we also included this for the analysis. We agree that adding the 'contrasts' suggested by the reviewer provides clarity.

In essence, our standard statistical approach is not complex, however, we accept that it may overwhelm a reader. To this end, we have now added the two main types of contrast to Figure 1 – with either SPECIES (6 levels, i.e., the respective plant species) or family (2 levels, i.e., Poaceae and Asteraceae) and highlight that these contrasts apply to both the previous and current generations of plants. We hope that this will clarify the statistics and thereby improve readability.

A couple of questions came up in my mind. What about Oomycetes, and what about nematodes? They have been shown to be extremely important in other PSF studies.

***Response: We agree with the reviewer that there are many components of the soil food web that may play a role in soil legacy effects. In the current study, we have not tested specifically for effects on oomycetes or nematodes, although expect that effects on nematode feeding groups would strongly relate to the effects on microbial groups (i.e. their food).

There are several reasons why we did not focus on nematodes (even though in various specific plant-soil systems they can act as important drivers of plant soil feedbacks, as indicated in e.g., Wilschut et al., 2019). First, nematode analyses, especially when repeated in time, require a relatively large volume of soil sample (ideally ~150 g or upwards), which would be difficult to collect without disturbing the mesocosms. As our main questions required a rather non-invasive methodology and sampling over time, this was not possible. Sampling for microbes, which requires only a few grams of soil is much less disturbing to the study system. Further, in other studies using the same plant species and soil type, we have performed various large scale nematode samplings e.g. in recent field experiments, but have not observed strong effects of for instance plant community composition on nematode communities (in fact, the only consistent finding is that nematode abundance increases over experimental time independent of treatment, see for instance Steinauer et al. 2020, *Ecosphere*).

Regarding Oomycetes and other protists, there is indeed some evidence that they can be involved in especially negative PSFs. There is also emerging evidence that oomycetes might be important inside the roots (Delavaux et al. 2021). However, we do regularly observe, across many other experiments in similar systems as the one used in the current manuscript, that plants drive bacterial and fungal communities (e.g. Hannula et al. 2019a,b; Heinen et al. 2018; 2020b), and that there are strong discrepancies between how bacteria and fungi in the soil behave over time (e.g. explored in great detail in Hannula et al. 2019b). Results of our recent field study on soil legacy effects revealed that it is predominantly soil fungal composition (as compared to bacterial composition or soil nutrients) that explains plant community responses to legacy effects. This led us to further explore these two particular (dominant) soil microbial groups in more detail.

Wilschut, R.A., van der Putten, W.H., Garbeva, P. *et al.* Root traits and belowground herbivores relate to plant–soil feedback variation among congeners. *Nat Commun* **10**, 1564 (2019). <https://doi.org/10.1038/s41467-019-09615-x>

Steinauer, K., Heinen, R., Hannula, S. E., De Long, J. R., Huberty, M., Jongen, R., ... & Bezemer, T. M. (2020). Above-belowground linkages of functionally dissimilar plant communities and soil properties in a grassland experiment. *Ecosphere*, *11*(9), e03246.

Delavaux, C.S., Schemanski, J.L., House, G.L. *et al.* Root pathogen diversity and composition varies with climate in undisturbed grasslands, but less so in anthropogenically disturbed grasslands. *ISME J* **15**, 304–317 (2021). <https://doi.org/10.1038/s41396-020-00783-z>

Hannula, S. E., Zhu, F., Heinen, R., & Bezemer, T. M. (2019a). Foliar-feeding insects acquire microbiomes from the soil rather than the host plant. *Nature communications*, 10(1), 1-9.

Hannula, S. E., Kielak, A. M., Steinauer, K., Huberty, M., Jongen, R., Jonathan, R., ... & Bezemer, T. M. (2019b). Time after time: temporal variation in the effects of grass and forb species on soil bacterial and fungal communities. *MBio*, 10(6).

Heinen, R., van der Sluijs, M., Biere, A., Harvey, J. A., & Bezemer, T. M. (2018). Plant community composition but not plant traits determine the outcome of soil legacy effects on plants and insects. *Journal of Ecology*, 106(3), 1217-1229.

Heinen, R., Hannula, S. E., De Long, J. R., Huberty, M., Jongen, R., Kielak, A., ... & Bezemer, T. M. (2020b). Plant community composition steers grassland vegetation via soil legacy effects. *Ecology letters*, 23(6), 973-982.

Also, monocultures rarely exist in nature except in highly managed agricultural situations so some discussion should be devoted to the more realistic scenario where multiple co-occurring plant species were present in the previous season and again in the current season? Also, the physical effects of soil are given less attention than biological effects. The spread of roots in the soil (depth especially) could also be an important factor where grasses tend to have shallower, fibrous root systems while some forbs have a deeper taproot. Admittedly, there is a lot of experimental power in the approach used.

***Response: The reviewer rightly points out that monocultures are rare in nature. (although it could be argued that monoculture patches in the dimensions of our containers, in the first, but definitely in the second phase, are not uncommon for the species that we studied in western European grasslands). In other work, that also focuses on the same plant species, we have examined how plant soil feedbacks operate in mixed plant communities, with realistic multispecies plant communities (Heinen et al. 2020). In the current study we selected a mechanistic approach, and to understand species-specific plant-soil-plant interactions, it is essential that these species are studied separately to avoid confounding effects of other co-occurring species

We agree with the reviewer that physical aspects of soils play an important role in plant performance. However, our research focuses on the biotic interactions occurring in the soil, and our recent findings have indicated that these may play a more important role in legacy effects than for instance plant mediated changes in soil nutrient composition in our system (which is not to say that it does not matter at all!). We also indicated this in the discussion section in L424-426 and 429-431.

Lastly, the reviewer points out differences between root systems of forbs and grasses. We fully agree, and have also regularly discussed these, for instance in Heinen et al. 2018; 2020a,b Hannula et al. 2019a;b; Steinauer et al. 2020. We agree that the points mentioned by the reviewer are likely to be a key cause for the differences in responses of these groups to soil legacy effects. In addition, it has been shown that grasses differ in terms of physical and chemical defenses from forbs, which we suspect to be another aspect that could play a role in mediating the creation of soil legacies, and the responses to those legacies (For instance discussed in Heinen et al. 2020a). We have now expanded our discussion to address these aspects of grasses vs. forbs and how they may explain the observed differences between the two functional groups (L353-355).

Heinen, R., van der Sluijs, M., Biere, A., Harvey, J. A., & Bezemer, T. M. (2018). Plant community composition but not plant traits determine the outcome of soil legacy effects on plants and insects. *Journal of Ecology*, 106(3), 1217-1229.

Heinen, R., Biere, A., & Bezemer, T. M. (2020a). Plant traits shape soil legacy effects on individual plant–insect interactions. *Oikos*, *129*(2), 261-273.

Heinen, R., Hannula, S. E., De Long, J. R., Huberty, M., Jongen, R., Kielak, A., ... & Bezemer, T. M. (2020b). Plant community composition steers grassland vegetation via soil legacy effects. *Ecology letters*, *23*(6), 973-982.

Hannula, S. E., Zhu, F., Heinen, R., & Bezemer, T. M. (2019a). Foliar-feeding insects acquire microbiomes from the soil rather than the host plant. *Nature communications*, *10*(1), 1-9.

Hannula, S. E., Kielak, A. M., Steinauer, K., Huberty, M., Jongen, R., Jonathan, R., ... & Bezemer, T. M. (2019b). Time after time: temporal variation in the effects of grass and forb species on soil bacterial and fungal communities. *MBio*, *10*(6).

Steinauer, K., Heinen, R., Hannula, S. E., De Long, J. R., Huberty, M., Jongen, R., ... & Bezemer, T. M. (2020). Above-belowground linkages of functionally dissimilar plant communities and soil properties in a grassland experiment. *Ecosphere*, *11*(9), e03246.

Was soil sampled for microbial testing after the first phase of growth with the 6 monocultures? See line 338, and Fig. 2. It seems that this is a black box (see Fig. 3, yellow bar). If there was no “starting point” quantification, some discussion of why is warranted. Also, why weren’t roots sampled over a time course instead of one time at end (L106). What are the origins of the soil used? Field soil? Was it sterilized before use? Also, say something about plants being AMF or not earlier.

***Response: Yes. In fact, it has been sequenced multiple times over the course of one year of conditioning. This has been described in detail in Hannula et al. 2019b. The ‘end point’ in this previous study marks the start of the second phase and the division into soil monoliths and subsequent planting with ‘current’ plants in this study happened shortly after. We use the information from the beginning community (then effect of current plant which becomes previous plant) as a starting point (like in figure 3, the yellow bar)

The reason why roots weren’t sampled is merely practical, as we would fully agree that this would have been an interesting parameter to include. Sampling root systems requires washing of the roots and, hence, a large disruption of the soil in the monoliths. For this reason, we have only measured root endophytic microbiomes at the destructive final sampling of the experiment. We have now made this more explicit in the methodology in L116, 118-120 and figure 1.

The origins of the soil have been described in detail in the first paragraph of the supplementary information and also elsewhere describing the first face of the experiment (Hannula et al. 2019b).

Further, the fact that all plants here form AMF is now mentioned in lines 69-50 of introduction and in material and methods on line 109. This is also discussed in discussion on lines 397-398.

No detailed data are presented on the microbiome results and the discussion of individual taxa (265-286, 358-375) seems like it is cherry-picking. Fig. 6 F as well. Additional information is presented in the Supplement and potentially some of it warrants presentation in the main text.

***Response: We have made a new figure (S6) to show all the possible correlations between endophytic bacterial phyla, fungal functional guilds and classes and root and shoot biomass. We show in the main figures that the whole community (measured with Bray-Curtis distance based community structure) affects the shoot and root biomass, and have highlighted the strongest correlations in the main figures. We agree that there are many other interesting interactions, but to keep the manuscript short and readable, we have not presented all these at once, but only the

strongest interactions in the main text and figures and therefore present the others in the supplementary information.

The design seems fundamentally nested where previous plants grouped by family and then species within family, but I am not sure that is how the statistics were done.

***Response: We thank the reviewer pointing this out. For the plant (shoot and root) data we re-ran the model taking both the identity of previous plant and the identity of the current plant into the model as random factors together with block. With this model we confirm that the family of the previous plant (Asteraceae/Poaceae) significantly affected the shoot biomass of the current plant ($F=6.77$, $p<0.05$) irrespective of the species of current or previous plant while the current plant family had no effect when current and previous plant species was taken as random factor. For the models on Asteraceae and Poaceae (Fig. 4A) we adjusted the statistics to include the previous and current plant identity in the model as random factors. This does not change the conclusion made based on the results presented in the figure: Poaceae show a significant growth reduction when grown in any Poaceae soil (new $F=20.12$, $p<0.001$ while Asteraceae are not affected by the family of the previous plant ($F=1.61$, $p=0.209$). The same analysis was performed for microbial NMDS data (Figure 3b) and phylum and classes of microbes.

We have specified the full models in the text now in L151-152 and show a graphic presentation of the models in Fig 1 now.

Specific comments

Part B and C in Fig. 3 are hard to follow.

***Response: We have increased the size of the individual panels. In principle, the thicker lines are those that are most important (the thin lines are individual replicate points, to show the variation), what can be clearly seen is that Asteraceae growing on Asteraceae soils and Poaceae growing on Poaceae soils maintain a similar microbiome (fungal) over time, irrespective of species. Poaceae growing in Asteraceae soils, and Asteraceae growing in Poaceae soils, after five months, become very similar (which is not to say that this may not change over time, but they did not diverge to become 'current Asteraceae' or 'current Poaceae' soils in five months). We feel that the pattern of the figure on fungi is much clearer than the one on bacteria, but this is also an important message; bacteria in soil do not respond to strongly to the family of plants.

L109-137 – sequencing and bioinformatics seem solid but not a lot of detail. Any references of past results to cite as verification?

***Response: References were now added in various places in methodology and supplementary methodology. Furthermore, we have revised the statistical methodology in various places for more clarity (and included an addition to Figure 1 for clarity of the applied approaches in the statistics). More details on the analysis can be found in the supplements and full R-code will be made available upon publication.

L150 – remind us of what the hypothesis is?

***Response: Done as suggested L156.

175 –not clear to me where three microbiomes comes from given two types cited later in sentence?

***Response: Two out of the four potential combinations of 2 previous and 2 current plant family combinations became very similar in fungal microbiome (or mycobiomes), that is, the Poaceae growing in soil with a legacy of Asteraceae, and the Asteraceae growing in soil with a legacy of Poaceae could not be differentiated from each other at the end of the experiment. This leaves us with three separate mycobiomes (all potential combinations being Asteraceae – Asteraceae, Asteraceae – Poaceae, Poaceae – Poaceae, and Poaceae – Asteraceae). We now clarified this sentence (L186-190).

Fig. 4 X axis hard to read – too small, which makes it very hard to interpret the data. Same comment for Fig. 6. And Fig. 7. Axes values need to be bigger and more legible. Even at 300% they were hard to see. It should be made clearer that the left column in B are grasses and the right are forbs.

***Response: We have adjusted axis size in all applicable figures. Furthermore, we now indicated functional group/family (Asteraceae and Poaceae) status in the two columns as suggested.

In Fig. 3 and 5, NMDS1 needs to be better explained.

***Response: We have added information on the first NMDS axis as a proxy of community structure used to simplify the figures. We have done the analysis on full model and use NMDS only for visual purposes, and included this in L147-148.

Fig. 7B is interesting but why the decline from 3 to 5 months?

***Response: We may speculate that feedback responses are strongest in early plant establishment, but that, as plants grow further/older, other aspects of the environment may start to play a comparatively stronger role diminishing the effect of microbes at later ages. Yet, the effect is strongest in the endophytic compartment. We have now incorporated this into the discussion in L407-408.

Reviewer #3 (Remarks to the Author):

The study design is neat and addresses an interesting aspect of plant microbial interactions in soil. Thus, the study has great potential but I cannot recommend that it is published in its current form. I must admit that I'm rather put off by the bold setting of the intro and a vagueness of statements that makes it difficult to assess how thoroughly the authors understand the broad topic addressed. For instance, the molecular methods applied are not usually very successful in detecting AM fungi and without discussing in any detail what level of detecting (nr of OTUs and their relative abundance) that is detected the authors go into interpretations of this group. This makes me very skeptical and I would like more convincing presentation in the main text. I do not have the expertise to evaluate the statistical analysis and I think the presentation of methods and results need to be more convincing at least for fungal biologist as myself. I have provided many details below that I hope the authors will find useful when revising the manuscript. Discussion needs to be streamlined, some arguments return several times and overall, it gets too speculative.
Comments throughout the text

***Response: We thank the reviewer for the very thorough and thoughtful review, and particularly the many specific comments that have helped us streamline the story.

We have now tried to streamline the story better to make this line of thought clearer across the manuscript and reworded our discussion on AMF. We now also included the caveats of sequencing soil microbiomes, and specifically pertaining to the conclusions that we can draw from it in the discussion (L389-401, especially line 391-392).

In the abstract the authors use the word wrought which is correct but I must admit that I did not know this word and had to look it up. Why not use the synonym shaped that is easily understood?

***Response: We thank the reviewer for pointing this out. As the reviewer concluded, it is a correct term in this context, but we are of course happy to follow this advice if it will make the manuscript more broadly understandable.

On line 14 the authors use the term long-term but they have previously on L 6 they said that this is about months. I do not see how several months can be considered a long-term effect if one think of soil processes or ecosystem processes, I think a more specific term or reference to the considered time span is preferable.

***Response: We have now revised the sentence to clarify our point. Our point was that plant-soil feedbacks are measurable as plants grow older (and therefore we used the term 'long-term'), but that our results suggest that an endophytic microbiome, taken up early in ontogeny, can play a role over a longer time period. We hope that this clarifies this point better.

The abstract ends with an interesting claim of being the firsts to actually show that plants shape soil microbiome. I agree that it is a commonly held assumption but maybe based only on indirect evidence because there is a multitude of studies showing that soil microbial communities as shaped by the plant communities (and vice versa). I see a problem with the present statement because it is very unspecific to say soil microbiome, what characteristic in the microbiome is it that is affected and reversible? I would like to see a more specific terminology such as community composition instead of microbiome.

***Response: We fear that the reviewer misinterpreted this sentence. Clearly, this is not the first study to show that plants shape the soil microbiome. However, this is (to our knowledge) the first that shows that soil microbial legacies shaped by previous plants can be reversed by a next generation of plants. This is one of the core assumptions of plant-soil feedbacks, but one that has not been tested empirically. Studies often take field soil, disregard pre-existing legacies, and start conditioning that soil for 10 weeks. We now show that, indeed, this shapes a new microbiome, but also that the pre-existing legacy may remain for much longer than that.

We have rephrased the final sentence of the abstract to better stress our point.

The introduction starts out in a rather generalized tone and lacks specificity. L 26: Why do the authors assume that these plant-mediated effects are restricted to the microbiome and can the effects of the microbiome really be separated from the direct effects of the litter in feed-back. This comes back on L 34 and it appears to me that the authors work under the assumption that feedback is solely a microbial process. Its surely refreshing with this attitude in contrast to many earlier studies that would assume that the feed back effect is all about litter composition and the chemical signature it leaves behind but maybe a more balanced view would be to at least in the introduction acknowledge that feed back is not just a microbial process.

***Response: We fully agree with the reviewer that plant-soil feedback, or soil legacy effects, are not purely a microbial process. It has been shown in countless studies that (as mentioned by the reviewer) litter input, allelopathic effects, changes in soil nutrient status, and probably various other factors all play a role in the effects of a plant via the soil on another plant. However, what we have observed, and what is the common consensus after three decades of plant-soil feedback research in grasslands, is that soil organisms can have a driving role in this process. We have in our group demonstrated various times that various soil organisms are affected by plants, and vice versa. In a series of recent papers (Heinen et al. 2020b, *Ecol Lett*; Hannula et al. 2019b, *mBio*), we have shown discrepancies between the effects of plants on soil bacterial and fungal communities, as well as the effect that these communities have on plants, ranging from individuals to plant communities in the field.

To answer the concerns of the reviewer, we have now made the first sentence more specific to grasslands. We would like to stress that there is ample evidence to support a driving role of microbes in plant-soil feedbacks in grasslands.

L 30: I don't know of any evidence that plant-soil feedbacks are general among functional groups such as grasses and forbs. Are those really examples of relevant functional groups with regard to plant feed back. I would expect early succession vs late succession plants to be more relevant and there are certainly examples of grasses and forbs in both these groups.

***Response: There is by now quite substantial evidence from different research groups that functional groups, such as grasses, forbs and legumes create different soil legacy effects, and that this is in part explained by soil microbial responses. In fact, plant functional group and ratios between functional groups were one of the strongest predictors in soil microbial legacy effects, as well as in plant community responses to soil legacies in a recent field study with the same plant species and soil type (Heinen et al. 2020b, *Ecol Lett*). Similar strong functional group effects were observed in other (greenhouse and common garden) studies by our group (e.g., Kos et al., 2015; Hannula et al. 2019a,b; Heinen et al. 2018; Heinen et al. 2020a) as well as by others (e.g., Petermann et al. 2008; De Kroon et al. 2012; Cortois et al. 2016).

We agree with the reviewer that microbial communities and resulting feedbacks of early and late successional species differ considerably, even directionally, (e.g., Morriën et al. 2017; Kardol et al.

2006).

Kos, M., Tuijl, M.A.B., de Roo, J., Mulder, P.P.J. and Bezemer, T.M. (2015), Species-specific plant–soil feedback effects on above-ground plant–insect interactions. *J Ecol*, 103: 904-914.

<https://doi.org/10.1111/1365-2745.12402>

Petermann, J. S., Fergus, A. J., Turnbull, L. A., & Schmid, B. (2008). Janzen-Connell effects are widespread and strong enough to maintain diversity in grasslands. *Ecology*, 89(9), 2399-2406.

de Kroon, H., Hendriks, M., van Ruijven, J., Ravenek, J., Padilla, F. M., Jongejans, E., ... & Mommer, L. (2012). Root responses to nutrients and soil biota: drivers of species coexistence and ecosystem productivity. *Journal of Ecology*, 100(1), 6-15.

Cortois, R., Schröder-Georgi, T., Weigelt, A., van der Putten, W. H., & De Deyn, G. B. (2016). Plant–soil feedbacks: role of plant functional group and plant traits. *Journal of Ecology*, 104(6), 1608-1617.

Morriën, E., Hannula, S. E., Snoek, L. B., Helmsing, N. R., Zweers, H., De Hollander, M., ... & Van Der Putten, W. H. (2017). Soil networks become more connected and take up more carbon as nature restoration progresses. *Nature communications*, 8(1), 1-10.

Kardol, P., Martijn Bezemer, T. and van der Putten, W.H. (2006), Temporal variation in plant–soil feedback controls succession. *Ecology Letters*, 9: 1080-1088. <https://doi.org/10.1111/j.1461-0248.2006.00953.x>

L 43-44. Its also very unlikely that an older plant would all of a sudden end up in a soil conditioned by another plant so it seems only logical that the effect of feed-back is stronger on seedlings.

***Response: We fully agree that plants are generally young when establishing in new environments. However, conceptually, it is important to understand how a plant responds to a soil legacy, and other work (e.g. Bezemer et al. 2018; Duddenhofer et al. 2018) has shown that older plants are less sensitive to soil mediated effects than young plants.

Bezemer, TM, Jing, J, Bakx-Schotman, JMT, Bijleveld, E-J. Plant competition alters the temporal dynamics of plant-soil feedbacks. *J Ecol*. 2018; 106: 2287– 2300. <https://doi.org/10.1111/1365-2745.12999>

Dudenhöffer, J-H, Ebeling, A, Klein, A-M, Wagg, C. Beyond biomass: Soil feedbacks are transient over plant life stages and alter fitness. *J Ecol*. 2018; 106: 230– 241. <https://doi.org/10.1111/1365-2745.12870>

L 51. I would also think that this is a founder effect, while older roots are already colonized there is no open niche to colonize so they are less susceptible to new colonizers.

***Response: It is certainly possible that a founder effect could occur and partly explains species occurrence in the roots. We would agree that likely these will buffer against new invaders, too. However, this does not mean that community composition within the root (i.e., relative abundance of species) of the founder communities will be stable over time, as microbial communities, especially those of bacteria, appear to be highly dynamic in time.

L58, I can not quite follow the sentence ending in soil origins? Please develop for clarity.

***Response: We have now changed the sentence to soils with a different history. This could be a different microbial composition, due to origin (location) but also due to other treatments. We hope that this clarifies the sentence (L59).

L 81. Im not convinced that selecting one family of forbs is a good rational to represent all forbs. It makes sense for comparison to grasses that is one family but there is no rational in what way Asteraceae represents all forbs. The presentation would be more convincing by being more accurate and less bold. Other than that I like the study design.

***Response: This is a fair point that we have seriously considered when designing the experiment. On the one hand, forbs are always more variable in their effects than grasses, because forbs represent such a diverse range of plant families, whereas most grasses fall in one, the Poaceae family. To compare with one family of grasses we selected one large family of forbs, that contains many common species in the types of grasslands we study. The Asteraceae are a dominant group of forbs in our study system, and the ones used here are among the most common ones in western European grasslands. Furthermore, the dichotomy between grass and forb effects also persist when a broader range of forb (i.e. in semi-natural communities is used), as we have observed in aforementioned studies.

We emphasized this in the original version, by mentioning this in the introduction and by including it also in Fig. 1. In essence, we compare two families, that represent two functional groups. We have now used the names of the two plant families, instead of functional groups throughout the text, but do briefly discuss aspects in the light of functional group as well.

Its not clear from the presentation in the intro if the entire plant was removed or if old root system is still present in the mesocosm. In natural soils we would expect roots to sit around but maybe decompose quite a bit from one season to the next. This is not clear in the methods either, see L 103 where it says removed.

***Response: The aboveground parts were removed, as were the plant bases leading into the ground. The roots were left to decompose, did not regrow, and we found very minimal remains when washing out roots at final harvest, indicated by fresh white living root systems for all but one species (notably *Festuca ovina* roots are generally brownish, even after washing).

We now made this clear in the methods (L112-L113).

L 90. Why did the authors hypothesis that its pathogens in the rhizosphere not in the roots that will affect plant growth?

***Response: Thanks for pointing this out. We have now changed this to 'endosphere and rhizosphere' (line 94), we agree that pathogens are expected to play a role in the soil and probably even more so in the root (covered in hypothesis iii).

L95, its very unspecific again to say that bacteria and fungi are different in growth strategies / traits. Fungi encompass both plant symbionts and plant pathogens to a much larger extent than bacteria and while there are some important plant pathogens among bacteria they are few compared to fungi

***Response: We agree, but this was not the point we intended to bring across. We want to say that bacteria have generally speaking higher turnover rates than fungi, and that this relates to their

growth rates / strategies. However, we do fully acknowledge that there are also single cell fungi with potentially higher turnover rates as well as hyphal forming bacteria. However, generally, evidence supports the notion that fungi have slower turnover rates and different traits. We have now revised this sentence in the text for clarity (L84-86)

Experimental design, its not clear from the text how many replicates the meso cosm was set up in.

***Response: Thanks for pointing this out, we had indicated this in Figure 1, but now realize that we had not indicated this in the text. We have added this to the methods for clarity (L105 and L114).

L 106 with the current sampling of roots do you assume that you sample the endosphere community or root associated community. It would be nice if this could be clarified.

***Response: Since roots were surface-sterilized following a rather rigorous earlier published and widely used protocol, including ethanol washing and ultrasonic bath treatments to dislodge remaining microbes adhered to the surface, we believe that what we have sequenced is predominantly what has been preserved within the root system. However, a complete removal of external DNA even after this rigorous protocol, cannot be fully ruled out. We have now included this in the revised version (L119-120).

L110 no empty line after heading level 2, or there should be an empty line after L 100 etc. Be consistent though out the text

***Response: Revised as suggested throughout the text.

L 121 no line break before this line

***Response: Thanks for pointing this out. Corrected as suggested.

L128, what confidene level was applied to accept an assigned taxonomy?

***Response: For bacteria we use ASVs and their identification was done using naive Bayesian classifier method (with selecting mimimum bootstrapping values) and clustering was done using 97% identity. Similarly, for fungi 97% clustering with UNITE was done. These are accepted methods and standard for both pipelines. For ASV/OTU with no reliable species assignment, higher level of classification was used.

L131 what kind of datasets are the authors referring to, sequences of known species? Unpublished information of life strategies?

***Response: Essentially FUNguild and equivalent databases combine sequence information to knowledge on ecology of species (based on cultured species and observational data). For example, plant pathologists have characterised almost all relevant (crop) pathogens and we could compare our sequences to the existing data to detect which of our sequence types are potentially pathogenic to plants.

L134, what component of the microbial community, all time points of soil community or only root community? Please be more specific

***Response: Thanks for pointing this out. In principle, we analysed relationships between community structure of soil and root microbiomes, both taken at five months (final harvest) and

plant growth. We made this explicit in the respective figures (5&6), but have now also clarified this in the methods in L146. Root microbiomes were very separate from the soil microbiomes and therefore we decided that analysing them together does not provide additional information (see figures S1 and S2).

L 135, here the authors use plant family rather than grass vs forbs. This is more accurate and should be used through out the text. Compare to L 147 where the hypothesis is referred to as being set up to test the difference between plant families.

***Response: Thanks for this suggestion. As explained above, we have debated this among ourselves thoroughly. On the one hand, we wanted to ensure a balanced design, but on the other hand, we do not want claims that are too broad. We now use plant family name throughout the text, but refer to both in the intro and discussion where this is relevant to the overall story line. We appreciate the reviewer's help in taking this decision.

L 152. What is the effect of root extension on how much of the soil is affected by the new plant. At what point in the experiment is the entire soil actually influenced by the plant? How densely planted are the systems and how much of the soil is affected by rhizodeposition. Also I cannot help to wonder what the effect of spatial variation is within each mesocosm. Not clear from the method description who sampling was designed to account for small scale spatial variation.

***Response: The species chosen all are relatively quick to root throughout the containers. Each rectangular container (13L), was planted with four equidistantly spaced seedlings (see picture in Fig. 1). It did not take long after the establishment for the aboveground biomass to fill the space. We can only speculate that the root compartment would have followed a similar pattern. We did observe that all species were very well rooted throughout the containers, with roots reaching every part of the soil volume in all species.

Regarding the soil sampling. Soil samples were taken at five positions in each container one next to each plant, and one in the center. These five samples were pooled and homogenized and used for sequencing, in the hope to reduce spatial effects, which we have now clarified in L117-118.

L 158-159 The spatial distribution is expected to vary between single celled bacteria and filamentous fungi, can this explain part of the difference in how community structure with new plant?

***Response: We agree. This may be part of the explanation. Bacteria and fungi are different in terms of growth rates and turnover rates, and this is related to their single-celled and multicellular lifestyles. We often find that bacterial legacies rapidly change, whereas those of fungi are much more stable over time (this temporal behaviour we have for instance described in detail in Hannula et al. 2019b, mBio). Clearly, (majority of) bacteria invest less in long life-spans than (most of) fungi, and this is what we see in this study as well.

L 161-162 this sentence makes no sense, previous and current had effect, just above it appeared to me that the effect was strongest from previous plant.

***Response: We agree that this sentence needed revision. The point we tried to make is that a proportion of the variance is explained by plant species (3 Poaceae, 3 Asteraceae), but irrespective of that, a separate analysis shows that there is also a strong 'family' effect (or, as we would also argue based on rigorous testing in a range of forb families previously, a functional group effect). The sentence has been revised for clarity (L162-164).

L 176 what does it mean to diverge and become similar?

***Response: Basically, we had four combinations of previous and current plant functional groups/families (all potential combinations being Asteraceae – Asteraceae, Asteraceae – Poaceae, Poaceae – Poaceae, and Poaceae – Asteraceae). There was an initial strong difference between all four, but two specifically changed to become more similar (Asteraceae - Poaceae and Poaceae - Asteraceae). We have now revised the sentence for clarity (L183-187).

Compare L 168 and L 178, should there be an indent or not?

***Response: These was checked and corrected.

L198 this does not quite make sense since 5 month is the only timepoint when there is biomass data so what other time point is there to compare link between biomass and soil microbial community. Consider re-phrasing to make clear what is being compared.

***Response: We compared the soil microbial community structure at all time points to the final biomass, to estimate at which time point the soil microbial community would explain most of the final plant biomass and show that for soil bacteria no time point was related to plant growth while for fungi the community structure at time of sampling was most related to biomass. We have now revised this for clarity (L209-215).

L231 given the molecular markers selected I'm surprised that the study identifies any AM fungi in the samples, the ITS2 regions is well known to work very poorly for these fungi and it would strengthen the statement if it included information on how many taxa were identified or how much of the root endophyte community belonged to AM fungi. Anyone working on AM is likely to be in doubt when reading this sentence. This links to L 250 where I would like to see a nr from x to y %. We know that different AM species may affect different plant species differently so I'm not convinced that a lower level of AMF is biologically important for this biomass in these experimental conditions.

***Response: These primers do amplify also AMF and in the ITS3mix primer one of the special modification is to amplify Glomeromycota (ITS3mix_4; Tedersoo et al. 2015). Furthermore it has been shown that especially within roots general fungal primers can be used to evaluate the relative abundance of AMF noting, however, that we cannot conclude on species present using ITS (Lekberg et al. 2018).

Here, besides changes in relative abundance of AMF in soils and roots, also the AMF community structure was influenced by the current and previous plants. However, as the species level assignment using ITS region is not optimal, but can be used to assess community structure, we report the major changes in community structure of AMF within roots, with the associated statistical analysis (Permanova for AMF inside roots; current plant: $R^2=0.18464$, $p<0.001$ and previous plant: $R^2=0.05725$ $p=0.002$). We have added this information to the supplementary figure S10 on AMF.

In total, there were 109 AMF taxa found in the system and they made a small portion (on average 3.3%) of the root endophytic fungi. The log transformed (due to non-normal distribution) relative abundances across time points and plants of AMF are presented in Fig S9. However, although their proportion is small, part of the change in total fungal community can certainly be attributed to changes in AMF which is something we also describe in the main text.

Lekberg, Y., Vasar, M., Bullington, L.S., Sepp, S.-K., Antunes, P.M., Bunn, R., Larkin, B. and Öpik, M. (2018), More bang for the buck? Can arbuscular mycorrhizal fungal communities be characterized

adequately alongside other fungi using general fungal primers?. *New Phytol*, 220: 971-976.
<https://doi.org/10.1111/nph.15035>

Tedersoo L, Anslan S, Bahram M, Põlme S, Riit T, Liiv I, Kõljalg U, Kisand V, Nilsson RH, Hildebrand F, Bork P, Abarenkov K (2015) Shotgun metagenomes and multiple primer pair-barcode combinations of amplicons reveal biases in metabarcoding analyses of fungi. *MycKeys* 10: 1-43.
<https://doi.org/10.3897/mycokeys.10.4852>

L252 only states an effect but not direction, this is not very useful for the reader. Same thing again on L 255

***Response: We have now indicated the increase in proportion of variance explained for these species as suggested (L253 and 257).

L265, the text has stated that grasses growing in own soil has more pathogens and then the order Agaricales is mentioned. It makes it sound like this is one of the examples of a pathogen, but surely the authors can not have meant this. Any taxonomy assignment above genus level has little relevance to assignment of functional group and Agaricales is a widely species rich group including many forest pathogens, I really hope this is now where they potential pathogens were identified. Again the main text needs more detail to avoid misunderstandings like this. Maybe just omitting the information on orders being more common. When is order a relevant ecological grouping anyway?

***Response: We understand where the confusion came from. Indeed, we did not mean to claim that Agaricales were all pathogenic. Instead, we meant to describe the family-specific effects of Poaceae and Asteraceae growing in 'own' relative to 'other' soils. The sentence on pathogens (main fig 7C), was followed by the effect on broader fungal orders (7D). Then we continued to discuss specific effects on potential pathogenic OTUs (supplementary material). We realize that it is better to describe the potential pathogens first, and then continue to the broader fungal orders in 7D. This should clear up the confusion (L281-284; L292-294).

L 272 observe that negative feedback loops are well known among AM fungi and their hosts. See extensive work by Bever cited in this paper (11, 12)

***Response: We are aware that such loops exist, and mention this in the discussion (L393-394).

L 303 again I have an objection to using long term in the context of months.

***Response: revised as suggested.

L307, both references are to 37. I'm not familiar with the conceptual idea that fungal growth rates are lower than bacterial, bacteria may have shorter reproduction but fungi likely forms larger biomass in soil, thus having larger spatial effects. I think it's also relevant to take into account how species rank curves look for fungi and bacteria. Fungal communities are usually dominated by a few dominant taxa, their effect thus may be larger than more even bacterial communities (if that's what bacterial communities look like).

***Response: We have corrected the reference. We did not measure the biomass of fungi and bacteria in this experiment but generally know that in these soils there are more bacteria than fungi (F:B ratio of around 1:5). We did check the rank abundance curves for (filtered) bacterial and fungal data for several of the samples and did not detect a difference in dominance of species between the

datasets. Hence, we think that this could be related to growth rates and not the dominance of certain species.

Figure rebuttal 1. Rank abundance curves of a number of samples for fungi and bacteria.

L316 I m thinking you could test this in your data by identifying what proportion of the community that is shared between roots and soil at harvest and the life strategies of those colonizing both roots and soil.

***Response: We have partly done these analysis but had not included this in the manuscript although we do find it an interesting question. We were concerned that the manuscript would contain too much data to condense into a coherent story. We have now added the community similarities between root microbiomes and soil micro biomes at different time points as supplementary figure S4 and mentioned this in results in section (L237-241)

Furthermore, for plant pathogenic taxa, soil pathogen abundance at time 1 and time 5 (but not at time 3) were correlated with each other across samples and quite some of the taxa were also shared between these time points. We can only hypothesize that these fungi enter the plant roots around t1 and are mature enough to grow already out of the plant in t5 which could explain the shared taxa in these time points.

L 332-333 taxonomic rank does not necessary reflect phylogenetic distance this these arguments are not valid unless you can also reference work comparing phylogeny and taxonomy for different plant families.

***Response: Due to the nature of our revisions, this comment is not applicable any longer. This is not mentioned in the discussion anymore. We now follow a plant family structure (Asteraceae vs. Poaceae), as suggested, throughout the manuscript.

L 343. I think this needs to be acknowledged already in the intro not to lose interest of the critical reader.

***Response: Thanks for pointing out. We have now acknowledged this in the introduction (L52-53).

L355, why was seed microbiome not examined if authors thought it might be very important. I think there are many references that would give good support for ruling out many of the observed root endophytes as being seed born.

***Response: We did add this because it would be a main mechanism for soil legacy effects but be rather explaining the microbiome of the 'current plant'. We do mention them as it may be one factor determining the endophytic compartment. We have not assessed this, because this is neither our field of expertise, nor will it answer our specific questions as we focus on the soil-mediated legacy effects of other plants. We have added sentences about 'current plant' microbiome coming partially from seeds now in introduction (L56-57) and in discussion (L363).

L376 I have not seen any information about AMF colonization. Detecting AMF reads is not the same as colonization which is inspected by staining of roots, if you had stained the roots that would be cool and I think it should be presented more clearly in the results.

***Response: We do agree that this would be very interesting. However, we did not analyse the percentage AMF colonization in the root samples. Arguably, this story is not about AMF colonization, but rather about microbial community structure. We do show that soil legacies do not affect the relative abundance of AMF in the roots so we decided not to pursue this line of research further. We have tried to streamline this in our manuscript, and have indicated the limitations this imposes for conclusions on AMF (L399-401).

We also point out that a recent study on N and P fertilization across grasslands showed an effect of fertilization on sequence based relative abundance of AMF but not on the colonization % which seems to be even more conservative measure (Lekberg et al. 2021) and hence here as we do not see a strong effect on relative abundance, we are likely also not see an effect on colonization.

Lekberg, Y., Arnillas, C.A., Borer, E.T. *et al.* Nitrogen and phosphorus fertilization consistently favor pathogenic over mutualistic fungi in grassland soils. *Nat Commun* **12**, 3484 (2021).
<https://doi.org/10.1038/s41467-021-23605-y>

L377 I think it's a sloppy style of writing to say: Plants with relatively fewer AMF ... is it OTUs assigned to AMF or reads representing these AMF. It is not fewer AMF its detection of reads. I would also be very cautious about interpreting relative abundance of AMF since they are notoriously difficult to detect in metabarcoding studies. Unless you can state specifically that you have a high and consistent detection you should be very careful.

***Response: We corrected this to 'relative abundance of AMF within plant roots. Indeed, it would have been informative to look into the AMF colonization within the roots but this was not in the scope of this study as we focused on microbial community composition responses. Although we are fairly confident due to high sequencing depth and consistent patterns across well replicated samples that the detected pattern holds, we agree that the proportion of AMF of the total fungal reads was rather low. We address that these results should thus be interpreted with caution (L399-401).

L379 on the contrary I would say that AMF species have been shown to have negative feedback and this is widely thought to be a reason for why AMF dominated plant communities are species rich.

***Response: We have revised this segment of the discussion and now indicate how AMFs may influence feedbacks (L393-396).

L394 I agree that you detect specific fungal pathogens in grass roots but you do not actually show that they in turn cause negative plant-soil feedback.

***Response: We agree that we cannot show that specific pathogens cause the reduction of growth but think that our results point towards a mechanism that grass species have pathogens causing a negative feedbacks on other grasses. We have rewritten these sentences. These pathogens occur at the same time as we detect a negative feedback. Yet future studies testing the exact mechanisms, particularly in the light of Koch's postulates (i.e., verify presence of specific microbe in affected plant individual, isolate microbe in pure culture, reproduce the effect upon addition of microbe to uninfected plant individual, recover microbe from infected plant individual) are needed.

L400 what exactly do you mean with plants show strong relationship?

***Response: This sentence was clarified.

Reviewer comments, second round -

Reviewer #1 (Remarks to the Author):

The authors have addressed all major issues raised by reviewers and done a fantastic job in improving the quality in the revised version. I have additional comment.

Reviewer #2 (Remarks to the Author):

Overall, I think that the revised manuscript is shorter and clearer, and that the main findings come through more strongly. As indicated in my previous review, there are many interesting and novel findings that will be of interest to the larger field of plant-soil feedbacks.

The author's response letter was very detailed and well-explained. Each of the three outside reviews was also very detailed. It is clear that the initial manuscript was evaluated very thoroughly, and revised in light of those comments. There were lots of references cited in the cover letter. It seems like the authors are out-sourcing their responses by citing these large numbers of studies (many repeatedly). It is not the duty of the reviewer to read a large body of outside literature before commenting on the focal manuscript, but I understand the point.

I believe that my earlier comments have been well addressed in this revision, along with the comments of the other reviewers. The experiments and hypotheses addressed are inherently complex and the authors have done a good job clarifying and simplifying their paper.

There are still several smaller points that could be addressed to strengthen to paper:

There is no 4 superscript in the author list so it is unclear who #4 refers to.

9, 12, 23-24 refer to plant, but it would be more accurate to say "plant species"

47 – define endophyte

63 – this assumes that the endophyte can propagate within the plant as it grows, vs. each occurrence representing an independent colonization event.

69 – set up

78 – did separation include below ground barriers? Otherwise roots and microbes could spread into other sections. 115 may clarify this question.

111 – clarify what n=5 refers to. Replicate monocultures per species?

190-193 – this is very interesting but could it be time dependent. After one year the mycobiomes might be quite different.

246-252 – This section seems like there should be either more information, or to eliminate it.

268-272 – This sentence is hard to follow.

294 and 295 – italicize species name

284-312 is a very long paragraph with many results and contrasts rolled in. It would be clearer if split into two or more smaller, more focused paragraphs.

395-396 – "inside plant roots and respective root" is unclear

Fig. 5A and 5B – the heading of soil fungi and soil bacteria could be modified to include "community"

Fig. 6E – typo on y axis label

Supplemental information – provides much detail on associations between microbial communities and plant growth characteristics.

Reviewer #3 (Remarks to the Author):

Thanks for a thorough revision of the manuscript and for the careful response to all my (sometimes ignorant) questions. I really appreciate the authors effort to provide a comprehensive response. The manuscript now reads very well and the added details makes for a solid impression and confident appreciation of the work. I have only some really minor things that should be addressed.

L284-586. The second part of the sentence does not make sense to me. If its about Asteraceae growing in Poaceae soil then there can be no independent of wheatear plants were grown in own soil, right? Because the plants that we are talking about here were all grown in soil of others.

A minor detail is that I think sentences should not start with a species name abbreviation, for example L278 and L 279 but in these cases the first species listed in the sentence should have its name written out.

L395 I think it should be root system here rather than root because it's the biomass of the whole root system that is used in the analysis, right?

L458 I think species is not the appropriate term here so rather than using ".." I think you could write taxa which is a term that nicely has the flexibility

The caption for figure 4 does not include a description of the two letter abbreviations that correspond to species names. I think this is needed and I wonder if you can could even illustrate in the figure which of these are Poaceae or Asteraceae.

Answers to the minor reviewer comments on manuscript:

Persistence of plant-mediated microbial soil legacy effects in soil and inside roots

We are very pleased to see that all reviewers were happy with the manuscript and have below answered the minor changes suggested. We have also gone through the editorial checklists and amended the manuscript based on comments there (like added versions to programs used). We further checked the legends of figures and added details to the descriptions when needed.

Reviewer #1 (Remarks to the Author):

The authors have addressed all major issues raised by reviewers and done a fantastic job in improving the quality in the revised version. I have additional comment.

Response: Thank you!

Reviewer #2 (Remarks to the Author):

Overall, I think that the revised manuscript is shorter and clearer, and that the main findings come through more strongly. As indicated in my previous review, there are many interesting and novel findings that will be of interest to the larger field of plant-soil feedbacks. The author's response letter was very detailed and well-explained. Each of the three outside reviews was also very detailed. It is clear that the initial manuscript was evaluated very thoroughly, and revised in light of those comments. There were lots of references cited in the cover letter. It seems like the authors are out-sourcing their responses by citing these large numbers of studies (many repeatedly). It is not the duty of the reviewer to read a large body of outside literature before commenting on the focal manuscript, but I understand the point. I believe that my earlier comments have been well addressed in this revision, along with the comments of the other reviewers. The experiments and hypotheses addressed are inherently complex and the authors have done a good job clarifying and simplifying their paper. There are still several smaller points that could be addressed to strengthen to paper:

Response: We thank the reviewer for positive remarks on the rebuttal and the current version of the manuscript.

There is no 4 superscript in the author list so it is unclear who #4 refers to.

Response: Corrected (refers to K. Steinauer)

9, 12, 23-24 refer to plant, but it would be more accurate to say "plant species"

Response: Corrected to read 'plant species'

47 – define endophyte

Response: added '(i.e. microbes living inside plant roots)'

63 – this assumes that the endophyte can propagate within the plant as it grows, vs. each occurrence representing an independent colonization event.

Response: We replaced the wording 'most' with 'many' to change the nuance of the sentence. Evidence shows that quite some are acquired early on and kept inside the roots but of course there are also others that are acquired later.

69 – set up

Response: corrected as suggested

78 – did separation include below ground barriers? Otherwise roots and microbes could spread into other sections.

Response: yes, they were plastic 'buckets' as illustrated in figure 1. We specify this now in line 77.

115 may clarify this question.

Response: indeed.

111 – clarify what n=5 refers to. Replicate monocultures per species?

Response: specified '(with 5 replicate monocultures per species)'

190-193 – this is very interesting but could it be time dependent. After one year the mycobiomes might be quite different.

Response: indeed, if the rate of change will stay the same, after one year of current plant growth, the legacy of previous plant in soil mycobiomes might be neglectable. We have added this in the discussion but more studies testing this are needed. We do show directional change for fungi as hypothesized in Fig. 2.

246-252 – This section seems like there should be either more information, or to eliminate it.

Response: we have decided to keep this part in the manuscript as this specifies which groups of endophytic bacteria and fungi are affected and we think this is relevant and interesting information to many.

268-272 – This sentence is hard to follow.

Response: We modified the sentence for clarity

294 and 295 – italicize species name

Response: Italicized names checked. Only species italicized while higher taxonomic levels of fungi not.

284-312 is a very long paragraph with many results and contrasts rolled in. It would be clearer if split into two or more smaller, more focused paragraphs.

Response: We divided this to two paragraphs, one on fungi and second one on bacteria and slightly shortened them.

395-396 – “inside plant roots and respective root” is unclear

Response: Added the word ‘root biomass of’ before ‘respective root’

Fig. 5A and 5B – the heading of soil fungi and soil bacteria could be modified to include “community”

Response: Changed to ‘soil fungal community’ and ‘soil bacterial community’

Fig. 6E – typo on y axis label

Response: Corrected.

Supplemental information – provides much detail on associations between microbial communities and plant growth characteristics.

Reviewer #3 (Remarks to the Author):

Thanks for a thorough revision of the manuscript and for the careful response to all my (sometimes ignorant) questions. I really appreciate the authors effort to provide a comprehensive response. The manuscript now reads very well and the added details makes for a solid impression and confident appreciation of the work. I have only some really minor things that should be addressed.

Response: We are happy to hear we have answered all the questions and that reviewer is pleased. Below we address the minor things further.

L284-586. The second part of the sentence does not make sense to me. If its about Asteraceae growing in Poaceae soil then there can be no independent of wheatear plants were grown in own soil, right? Because the plants that we are talking about here were all grown in soil of others.

Response: Indeed, this was unclear. We changed this to: ‘in roots of all Poaceae plants growing in their respective own soils’. The model was also corrected for the plant identity.

A minor detail is that I think sentences should not start with a species name abbreviation, for example L278 and L 279 but in these cases the first species listed in the sentence should have its name written out.

Response: Corrected to read '*Alopecurus pratensis*'

L395 I think it should be root system here rather than root because it's the biomass of the whole root system that is used in the analysis, right?

Response: Added the word 'system'

L458 I think species is not the appropriate term here so rather than using ".." I think you could write taxa which is a term that nicely has the flexibility

Response: Changed as suggested, indeed better to use the word 'taxa'

The caption for figure 4 does not include a description of the two letter abbreviations that correspond to species names. I think this is needed and I wonder if you can could even illustrate in the figure which of these are Poaceae or Asteraceae.

Response: We have added the species names behind abbreviations to the legend and mention that green colors correspond to grasses and blue colors to forbs.